# A role for ColV plasmids in the evolution of pathogenic *Escherichia coli* ST58

Cameron J. Reid [1✉], Max L. Cummins [1], Stefan Börjesson [2,3], Michael S. M. Brouwer[4], Henrik Hasman[5], Anette M. Hammerum[5], Louise Roer[5], Stefanie Hess[6], Thomas Berendonk[7], Kristina Nešporová [8,9], Marisa Haenni[10], Jean-Yves Madec[10], Astrid Bethe [11,12], Geovana B. Michael[11,12], Anne-Kathrin Schink[11,12], Stefan Schwarz [11,12], Monika Dolejska[8,9,13] & Steven P. Djordjevic [1✉]

*Escherichia coli* ST58 has recently emerged as a globally disseminated uropathogen that often progresses to sepsis. Unlike most pandemic extra-intestinal pathogenic *E. coli* (ExPEC), which belong to pathogenic phylogroup B2, ST58 belongs to the environmental/commensal phylogroup B1. Here, we present a pan-genomic analysis of a global collection of 752 ST58 isolates from diverse sources. We identify a large ST58 sub-lineage characterized by near ubiquitous carriage of ColV plasmids, which carry genes encoding virulence factors, and by a distinct accessory genome including genes typical of the Yersiniabactin High Pathogenicity Island. This sub-lineage includes three-quarters of all ExPEC sequences in our study and has a broad host range, although poultry and porcine sources predominate. By contrast, strains isolated from cattle often lack ColV plasmids. Our data indicate that ColV plasmid acquisition contributed to the divergence of the major ST58 sub-lineage, and different sub-lineages inhabit poultry, swine and cattle.

[1] iThree Institute, University of Technology Sydney, Ultimo, NSW 2007, Australia. [2] Department of Animal Health and Antimicrobial Strategies, National Veterinary Institute (SVA), 75189 Uppsala, Sweden. [3] Department of Microbiology, Public Health Agency of Sweden, 17182 Solna, Sweden. [4] Wageningen Bioveterinary Research, Lelystad, Netherlands. [5] Department of Bacteria, Parasites and Fungi, Statens Serum Institut, Copenhagen S, Denmark. [6] Institute of Microbiology, Technische Universität Dresden, Dresden, Germany. [7] Institute of Hydrobiology, Technische Universität Dresden, Dresden, Germany. [8] CEITEC VETUNI, University of Veterinary Sciences Brno, Brno, Czech Republic. [9] Department of Biology and Wildlife Disease, Faculty of Veterinary Hygiene and Ecology, University of Veterinary Sciences Brno, Brno, Czech Republic. [10] Université de Lyon-ANSES, Unité Antibiorésistance et Virulence Bactériennes, Lyon, France. [11] Institute of Microbiology and Epizootics, Centre for Infection Medicine, Department of Veterinary Medicine, Freie Universität Berlin, Berlin, Germany. [12] Veterinary Centre for Resistance Research (TZR), Freie Universität Berlin, 14163 Berlin, Germany. [13] Biomedical Center, Charles University, Charles, Czech Republic. ✉email: Cameron.Reid@uts.edu.au; Steven.Djordjevic@uts.edu.au

*E*scherichia coli predominantly live as harmless commensals in the gastrointestinal tract of mammals and birds. They also reside, independent of a host, in environmental habitats including water, soil and sediments. If they possess or acquire factors that allow them to adapt to niches in humans that are outside the gut, they can cause disease. The *E. coli* pathotype that accounts for the vast majority of human extra-intestinal pathologies—urinary tract infections, pyelonephritis, sepsis and meningitis—is known as extra-intestinal pathogenic *E. coli* (ExPEC). The pathologies it causes place a significant burden on health systems globally[1].

ST58 (clonal complex (CC) 155) is a persistent ExPEC clonal group that is unusual among ExPEC strains by belonging to phylogroup B1, one of eight stable phylogroups used to classify *E. coli*. Most ExPEC strains, including those in the globally dominant ST131 clonal group, are members of phylogroup B2[2]. The emergence of ST58 is perhaps best highlighted by a recent study that found the proportion of B1-ST58 isolated from bloodstream infections in the Paris region has more than doubled over a 12 year period in contrast to stable proportions of infections caused by phylogroup B2 isolates[3]. Beyond this study, *E. coli* ST58 is increasingly responsible for both sporadic and persistent cases of bloodstream infections in humans across the globe[4–12]. As well as colonising humans, ST58 has been identified in healthy and diseased food-producing animals (cattle, poultry, swine) and in poultry farm-associated flies, manure and water[8,13–21]. Early reports of ST58 postulated a wild animal source and it has since been reported in commensal and pathogenic cases from wild, captive and companion animals, with wild birds from both urban and pristine environments emerging globally as an important source[22–33]. Food sources of ST58 include chicken and turkey meat, barley and oats, raw meat-based pet food and store-bought produce[8,34–37]. Further concern arises from widespread observations of ST58 carrying antimicrobial resistance genes (ARGs). Human faecal isolates of ST58 from healthy individuals in Tunisia, Sweden and the Netherlands[38–41] have been found harbouring genes conferring resistance to third-generation antimicrobials, as have ST58 isolates from food-producing animals[8,13–21] ($bla_{CTX-M}$ genes encoding for extended-spectrum beta-lactamase (ESBL) production). Multidrug-resistant (MDR) and ESBL-producing ST58 also contaminate soil, rivers, mangroves and wastewater[35,42–44].

We have limited knowledge of how ST58 emerged as a human pathogen. How did it come to evolve within phylogroup B1, which is rarely pathogenic?[2] Though host factors are the greatest predictor of extra-intestinal virulence, intrinsic factors include adhesins, toxins, protectins and iron acquisition systems[45]. Phage and plasmid mobilised iron acquisition systems in particular, including yersiniabactin (HPI) and aerobactin among others, have been shown to play a major role in intrinsic virulence across the genus *Escherichia*[46]. The majority of existing studies reporting ST58, and indeed most studies on ExPEC, are limited by their employment of ESBL-selection criteria. Although clinically coherent, this selection criteria ignores the ecological complexity of AMR and pathogen emergence. The consequences are significant—ESBL-selection criteria drive misleading generalisations about the AMR status of clonal groups and actively prevent a complete understanding of their evolutionary history, particularly with regard to the identification of non-AMR traits that play significant roles in fitness and pathogenesis[47].

We recently reported an ST58 strain carrying a ColV plasmid that caused urosepsis in a Sydney Hospital[4]. ColV plasmids are a subset of typically conjugative F plasmids abundant in poultry and have also been reported in healthy human faecal commensal *E. coli* and ExPEC[48]. They have known pathogenic properties and traits involved in intestinal fitness[4,49–53], and carry a variety of

ARGs, class 1 integrons and mercury resistance transposons[48]. Therefore, there are numerous selective pressures that may contribute to their persistence in animal and human hosts. ColV plasmids are carried at high levels in important human pathogens with poultry associations including ExPEC clonal groups ST95 and ST117, though their relative abundance in other STs is unknown[54].

In this work, we perform a pan-genomic epidemiological analysis of a global collection of 752 ST58 isolated from humans, animals and environmental sources. Hypothesising that ColV plasmids had a role in the emergence of ST58 as a pathogen, we query their presence and source distribution in a collection of 34,364 draft *E. coli* genome assemblies from Enterobase. Our findings support the role of ColV plasmids and non-human sources in the evolution of pathogenic ST58 as well as the idea that pathogen emergence should be understood within a One Health framework, resulting from a complex of mechanistic drivers and networked pathways between environments and hosts rather than discretely acting selective pressures.

## Results

**A diverse collection of ST58 genomes**. We first built a collection of *E. coli* ST58 genome sequences from isolates that were diverse in origin—temporally, by source and geographically (Fig. 1(a–c)). In total, our genome collection comprised 752 whole-genome sequences of *E. coli* ST58 isolates collected between 1970 and 2019 from 6 niches sub-divided into 15 sources, Fig. 1b) across 33 countries on 6 continents (Supplementary Data 1). Of the 752 sequences, 178 were sourced from in-house and collaborator collections and the remainder from Enterobase. Within the collection, 92% of sequences (692/752) were from isolates collected after the year 2000 (See Fig. S1 for full range). The most common niche represented in the collection was livestock, accounting for 68% of the collection (514/752). Livestock was followed distantly by wild animals (12%, 93/752) and humans (11%, 79/752). Livestock isolates were predominantly from bovine (35%, 262/752), poultry (17%, 125/752) and porcine sources (14%, 106/752). Among wild animals isolates, 11% were from avian sources (84/752). More than half of the human isolates were from people suffering from extra-intestinal *Escherichia coli* pathologies (ExPEC; 6% of the total collection, 43/752). The most represented continents in the collection were North America (60%, 454/752) and Europe (25%, 189/752).

**E. coli ST58 contains a major sub-lineage rich in ColV virulence plasmids**. To investigate evolutionary relationships within our collection of ST58 genomic sequences, we first generated a global ST58 phylogeny inferred from the core gene multi-alignment.

The core gene phylogeny was grouped into six major clusters by fastbaps (designated BAP1-6; See Methods), encompassing 750 sequences. The outgroup strain, belonging to ST155, and two ST58 strains on divergent branches near the root comprised BAP clusters 7, 8 and 9, which are hereafter excluded from cluster-based analyses. Each major cluster was characterised by source, serotype, *fimH* allele, F plasmid replicon sequence type (F RSTs) and inference of ColV plasmid presence. Two of the clusters included the vast majority of collection sequences: BAP6 ($n = 205$) and BAP2 ($n = 363$), (Fig. 2). BAP6 sequences were more commonly from bovine than other sources (bovine: 142/205, 69%) but they displayed a diversity of F plasmid RSTs, serotypes and *fimH* alleles (Figs. 3, S2, S3). In contrast, BAP2 exhibited a greater distribution of sources and dominance of specific serotypes and *fimH* alleles (Fig. 3). The most striking feature of the BAP2 cluster was the high proportion of sequences

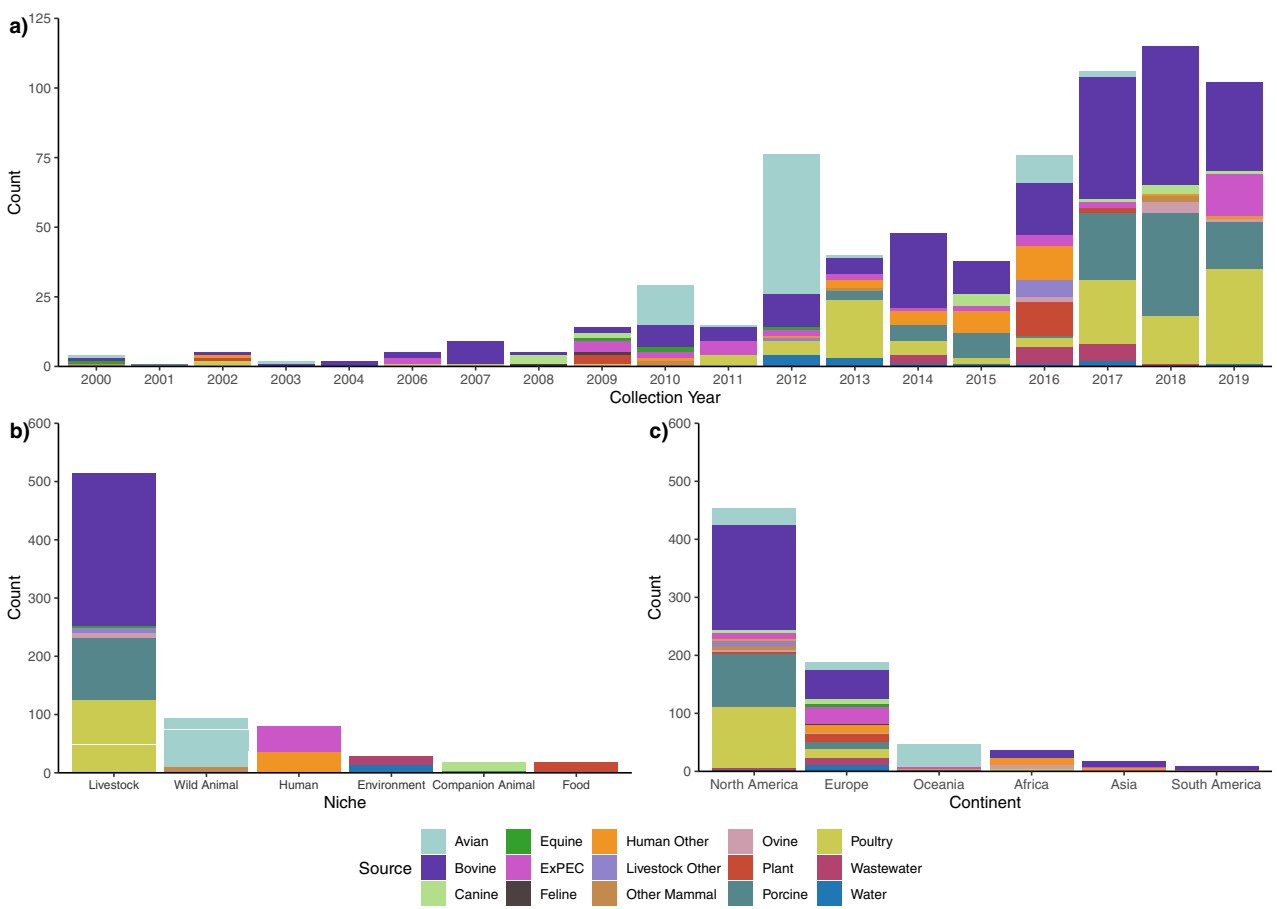

**Fig. 1 ST58 genome collection metadata.** Summary of metadata of 752 *E. coli* ST58 strains stratified by source. **a** Collection year (2000–2019; for full year distribution see Fig. S1); **b** Niche; **c** Continent.

that featured ColV F plasmids (308/363, 85%). These plasmids were represented predominantly by three RSTs: F2:B1, F18:A6:B1 and F18:B1 (Figs. 3b, S4, Supplementary Data 2).

The initial screening of the cluster sequences for the presence of a ColV plasmid was based on the Liu criteria (see Methods, Fig. S5). However, as the Liu criteria are limited by screening for specific genes without consideration of plasmid backbone structure, we sought to corroborate the identification of ColV plasmids via an additional methodology. For this purpose, we aligned de novo assemblies to the backbone of archetypal F2:B1 ColV plasmid pCERC4 and visualised the alignment as a heatmap of binned nucleotide identities (Fig. 4). Overall, results obtained using each methodology corresponded well. Coverage patterns of ColV-positive (ColV+) sequences outside the BAP2 cluster were similar to those within it.

Most of the ColV+ genomes that were identified in our collection originated from sources associated with food production (poultry, porcine, bovine) and extra-intestinal disease in humans (ExPEC) (Fig. 5, Fig. S6). For three of these sources, the vast majority of genome sequences were ColV+: poultry (109/125, 87%), porcine (77/106, 73%) and ExPEC (33/43, 77%). In contrast, the proportion of bovine sequences that were ColV+ was notably less (48/262, 18%).

**The BAP2 cluster displays a distinctive accessory genome.** Having identified at the core genome level a distinct ColV+ sublineage within ST58—the BAP2 cluster—we hypothesised that it would also exhibit fundamental differences in its accessory

genome relative to the remainder of the phylogeny. To test this we performed a pangenome-wide association study (pan-GWAS), using BAP2 cluster membership as the test variable. Within the pangenome, 78 genes coding for non-hypothetical proteins were over-represented in the BAP2 cluster and 55 were under-represented (these genes are therefore associated with non-BAP2 ST58). We also performed this analysis for the other BAP groups; however, only BAP6 contained genes that met our significance threshold (1E-50) for over- or under-representation, and these *p*-values were generally far lower than any of the genes associated with BAP2 (Fig. S7, Supplementary Data 3). Mapping genes identified in the BAP2 analysis back to the core gene phylogeny revealed that a sub-group of approximately 294 genomes within BAP2 contained a highly conserved accessory gene profile (Fig. 6). In addition to the expected identification of genes present on ColV plasmids, several other genes stood out in the BAP2-associated accessory gene profile. The two most highly associated genes in the cluster, *ugd* and *galF*, both encode enzymes involved in outer membrane lipopolysaccharide biosynthesis. Two prophage integrase *intA* genes were associated with the clade, one of which was exclusively associated with the aforementioned 294 sequence sub-group in the BAP2 cluster. *mlrA*, a regulator of curli biosynthesis and biofilm formation was also exclusively associated with this sub-group. *fyuA*, a marker gene for Yersiniabactin High Pathogenicity Island (HPI), which encodes an iron uptake system that is a major contributor to intrinsic extra-intestinal virulence across the genus *Escherichia*, was also associated with BAP2 indicating that most of these ST58 have acquired an HPI-like pathogenicity island. This was

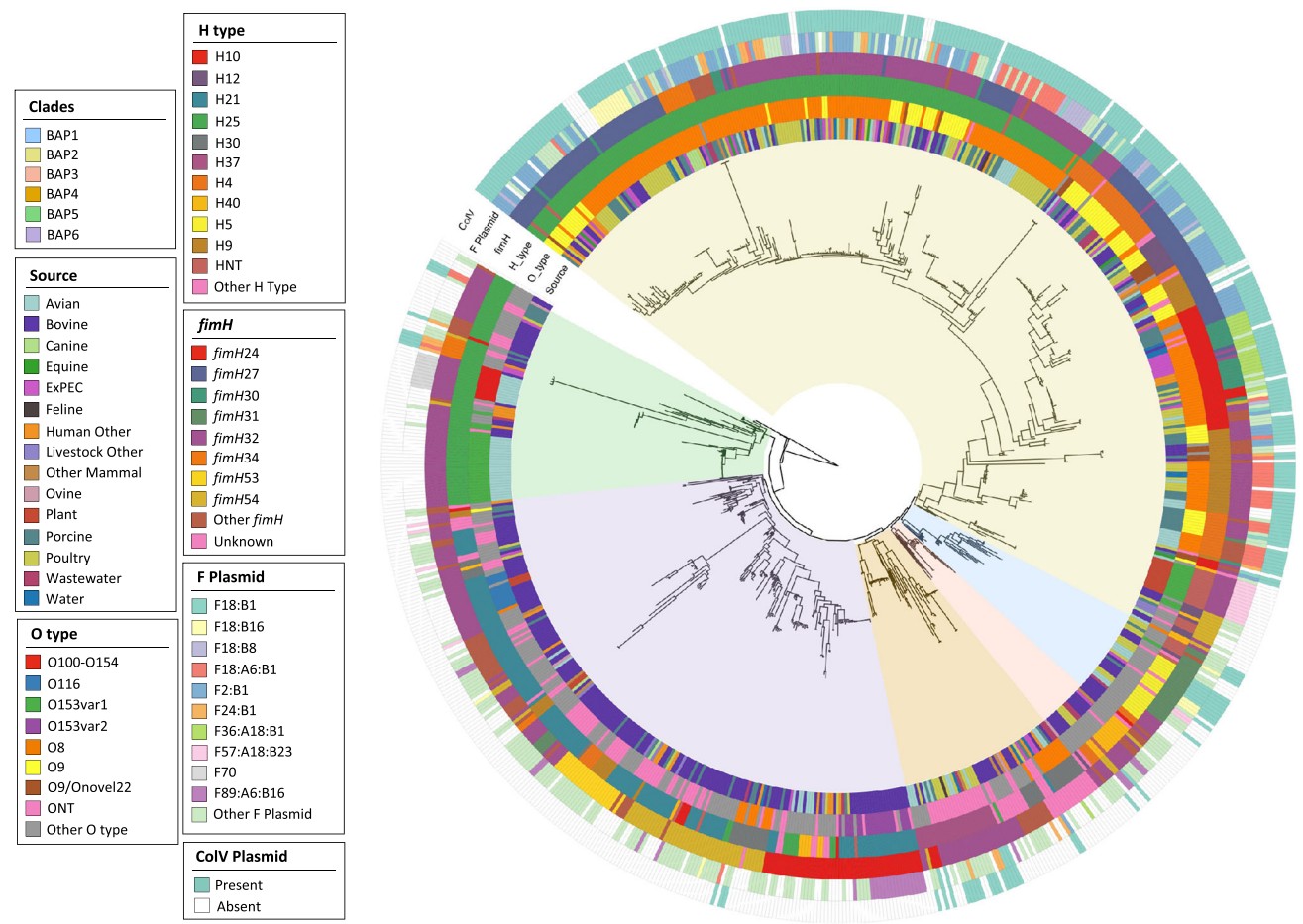

**Fig. 2 Phylogenetic tree with metadata.** Maximum-likelihood core gene phylogeny of 752 *E. coli* ST58 rooted on ST155 strain MOD-EC5019 with genotypic information and metadata.

supported by virulence gene screening which identified both *fyuA* (306/363; 84%) and cognate HPI marker, *irp2* (303/363; 83%), in a high proportion of BAP2 sequences (Fig. S9).

Twenty-seven genes were present in both BAP2 over- and under-represented groups, indicative of gene families that are likely core to ST58 but have alternative sequences in BAP2 compared to the remainder of the phylogeny (Supplementary Data 3). Most of these paralogous genes were involved in metabolism, membrane transport processes and DNA transcription, notably including genes of the *bcs* operon involved in cellulose metabolism, an abundant polysaccharide in the bovine rumen. These results suggest that the BAP2 cluster likely displays a number of functional differences to the remainder of ST58 in relation to multiple biological processes.

Overall the pan-GWAS analysis implies that both sequence divergence in core ST58 genes and multiple instances of horizontal gene transfer events involving plasmids, phages and genomic islands, have contributed to the evolution of the BAP2 cluster.

**ColV+ and BAP2 genomes carry more antimicrobial resistance genes and virulence genes**. We screened our collection of ST58 *E. coli* genomes for antimicrobial resistance genes (ARGs) and virulence-associated genes (VAGs). Strain-wise totals for each were compared with respect to BAP cluster membership and ColV status. This analysis revealed that, BAP2 sequences carried significantly more ARGs on average than the other clusters except BAP1 (Pairwise Wilcoxon test with Benjamini–Hochberg

adjusted p-values: vs BAP3 $p = 8.63e-8$; BAP4 $p = 0.036$; BAP5 $p = 6.26e-6$; BAP6 $p = 1.38e-28$), and more VAGs than all other clusters (Pairwise Wilcoxon test with Benjamini–Hochberg adjusted p-values: vs BAP1 $p = 5.23e-5$; BAP3 $p = 3.1e-9$; BAP4 $p = 2.72e-16$; BAP5 $p1.67e-21$; BAP6 $p = 5.96e-70$) (Fig. 7a, b). Similarly, ColV+ strains carry more ARGs (Two-sided Wilcoxon Rank-sum test: $p = 9.2e-34$) and VAGs (Two-sided Wilcoxon Rank-sum test: $p = 1.28e-112$) than ColV- strains (Fig. 7c, d).

A wide variety of ARGs were identified in the total collection including genes conferring resistance to older antimicrobial compounds such as ampicillin ($bla_{TEM-1B}$; 248; 33%), streptomycin ($strAB$; 296; 39% and 294; 39%), sulphonamides ($sul2$; 279; 37%), tetracyclines ($tet$(A); 272; 36% and $tet$(B); 165; 22%) and trimethoprim ($dfrA5$; 127; 17%). Genes mediating resistance to third-generation cephalosporins and carbapenems were comparatively less common. ESBL-encoding genes included $bla_{CTX-M-1}$ (49; 7%), $bla_{CTX-M-55}$ (23; 3%) and $bla_{CTX-M-15}$ (21; 3%). AmpC beta-lactamase $bla_{CMY-2}$ (61; 8%) was also identified. The class 1 integron-integrase gene, *intI* was present in 245 sequences (33%). Integrons are a major driver of bacterial evolution and *intI* is commonly linked with ARGs. Mercury resistance gene *merA* was present in 22% of sequences (164/752). Point mutations associated with resistance to fluoroquinolones were only identified in 9% of all sequences and 12% of BAP2 sequences (Fig. S8).

Nearly all the most common VAGs in the full collection (those present in more than 200 sequences) were concentrated within the BAP2 cluster and known to be carried on ColV plasmids. Abundant non-ColV VAGs included those encoding increased

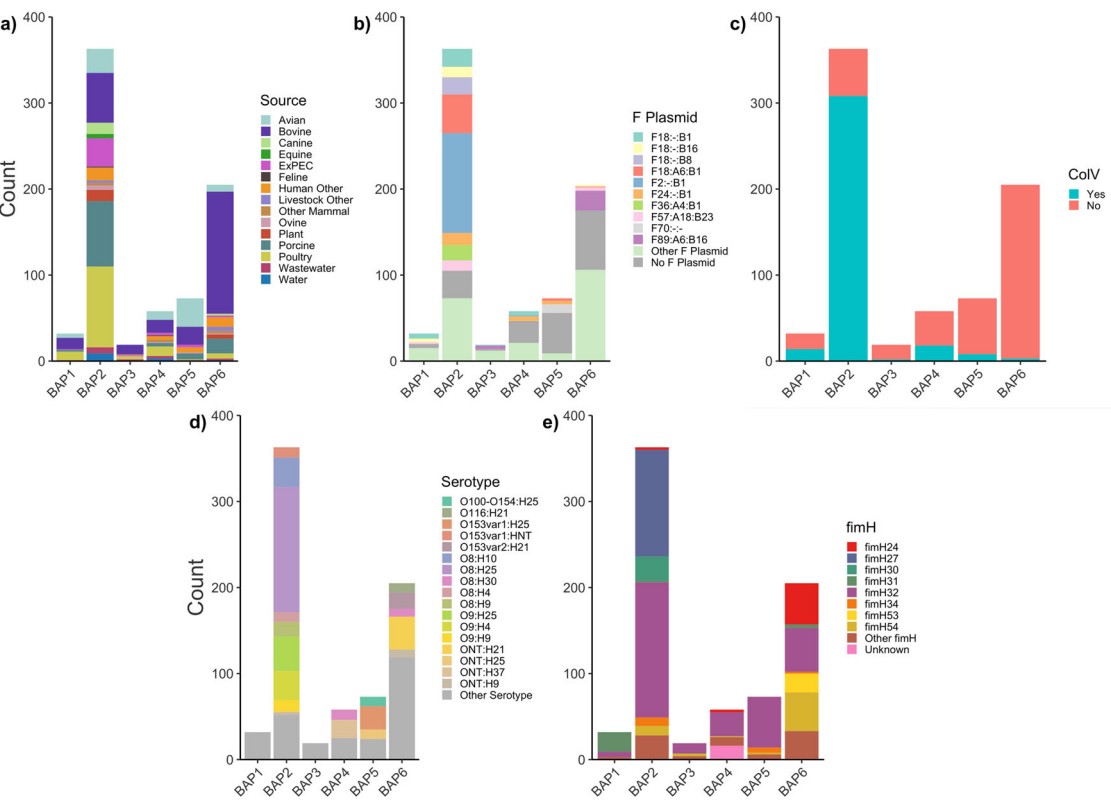

**Fig. 3 Metadata distribution across BAP clusters.** Distribution of **a** source; **b** F plasmid replicon sequence type (RST); **c** ColV carriage; **d** serotype and **e** *fimH* allele by BAP cluster for 750 ST58 genome sequences.

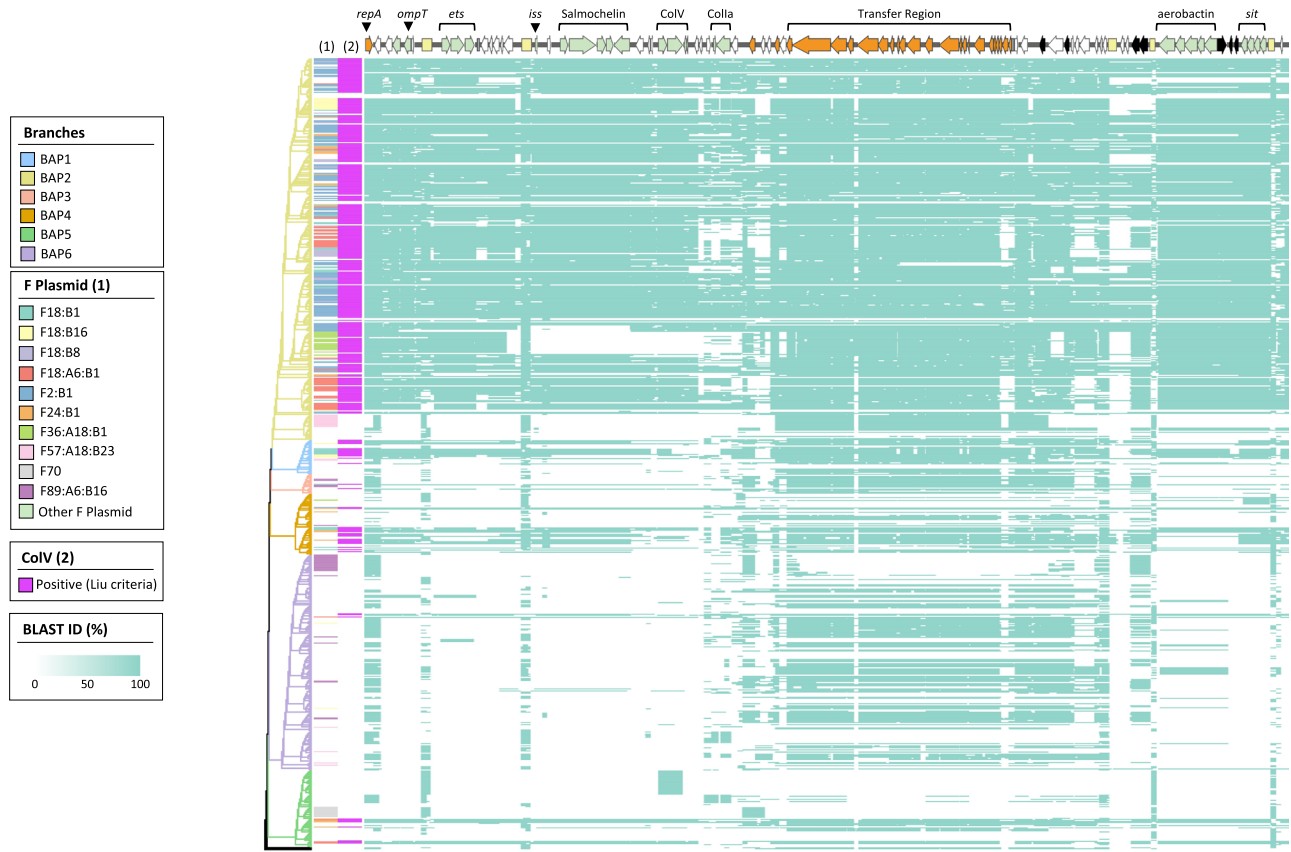

**Fig. 4 Alignment of ST58 sequences to ColV plasmid pCERC4.** Heatmap of nucleotide identity across 100 bp segments of the pCERC4 backbone. Tree branches are coloured by BAP cluster. F plasmid replicon sequence type (RST) and ColV presence are indicated in panels (1) and (2).

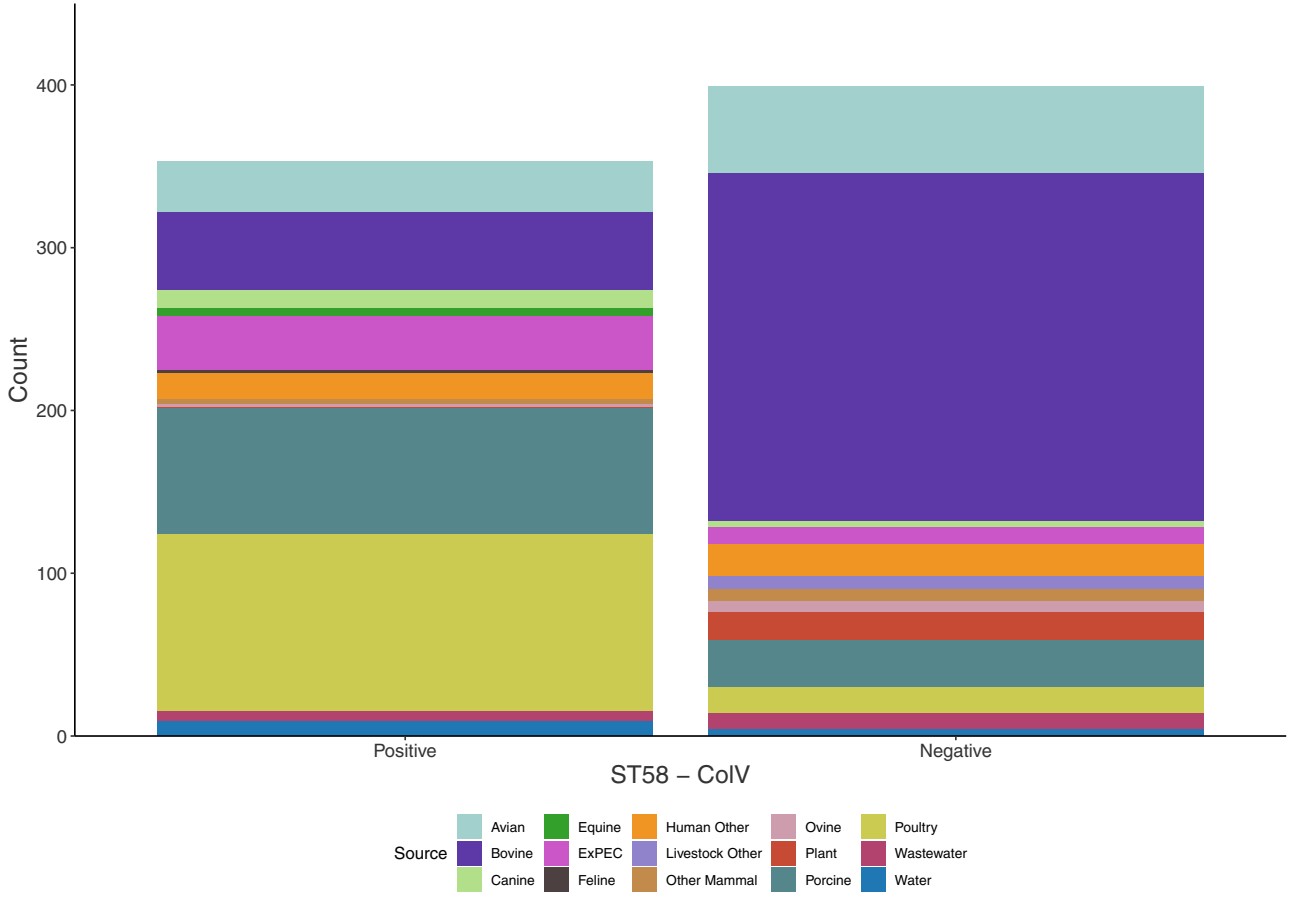

**Fig. 5 ColV carriage by source.** ColV carriage stratified by source in the ST58 genome collection.

serum survival protein *iss* (583; 78%; note: *iss* is frequently observed on ColV plasmids but is not included in the marker gene criteria); colonisation factor and fimbrial adhesin *fimH* (730; 97%); ferric dicitrate receptor *fecA* (412; 55%); high pathogenicity island marker and yersiniabactin iron receptor *fyuA* (347; 46%), and bacterial defence factor microcin transport protein *mchF* (324; 43%). Toxin carriage was rare though enterohaemolytic *E. coli* (EHEC) marker genes, including *stx* variants and *subA*, respectively, were identified in some strains (Fig. S9).

Aside from F plasmids, plasmids from incompatibility group IncI1 were the most commonly identified (228/752, 30%). IncI1 plasmids were present in all major clusters, but were diverse by IncI1 pMLST. (Fig. S10). Sixty-seven sequences that contained an IncI1 replicon did not have an identifiable pMLST. Isolates sourced from poultry had the highest intra-source proportion of IncI1-positive sequences (67/125; 53.6%).

**The high ColV carriage rate in ST58 is comparable to other emerging ExPEC with major reservoirs in food animals.** To date, most data on the presence of ColV plasmids in *E. coli* has been obtained from avian or human ExPEC isolates. In addition to these sources, we have observed, in high proportions, ColV plasmids in ST58 BAP2 genomes originating from other food production animals, particularly pigs. Thus, we wanted to investigate how common it was for ColV plasmids to be present in the genomes of *E. coli* isolated from other sources, as well as the range of *E. coli* STs hosting ColV plasmids. To do this, we curated and analysed a collection of 34,364 draft *E. coli* genome assemblies from Enterobase (Supplementary Data 4). We found that within this collection, poultry (2328/4260, 55%) was by far

the most dominant source for ColV+ *E. coli*, followed by porcine (526/2683, 20%) and human ExPEC (735/4465, 16%; Fig. 8a). Low carriage was observed in human other (523/15088; 3%) and bovine sources (176/6926; 3%). The rate of ColV carriage in all genomes was 13% (4370/34,364; Fig. 8b). ColV carriage rates summarised by ST and Source were compared for fifteen STs containing more than 100 assemblies and ColV carriage rate greater than 10% (Fig. 8c, d). Whilst the methodology for this collection of sequences captured only 588 ST58 draft genome assemblies (most of which are present in the primary collection under analysis in this manuscript), the ColV carriage rate of 48% (281/588) was only slightly higher than the estimate for the primary collection at 46% (353/752). Only five STs (G-ST117, C-ST88, B1-ST162, A-ST93 and C-ST23) had higher ColV carriage rates than ST58. Four of these STs belong to the 'environmental' phylogroups A, B1 and C whilst globally dominant APEC ST117 now belongs to phylogroup G, having previously been considered D or F[55]. Poultry-sourced sequences dominated these top six STs though distributions varied, with ST88 displaying more porcine sequences. ST95, ST73 and ST12, all of which belong to 'pathogenic' phylogroup B2 contrastingly displayed mostly human sources, either ExPEC or other. Almost all of the STs present feature in numerous reports of human or animal infection, or in conjunction with concerning AMR genotypes.

**Genomic linkages between human and non-human source ST58.** We hypothesised that a proportion of ST58 that cause human extra-intestinal infections are closely related to ST58 found in non-human sources. To test this, we determined pairwise SNP counts between all sequences (Fig. S11) and then

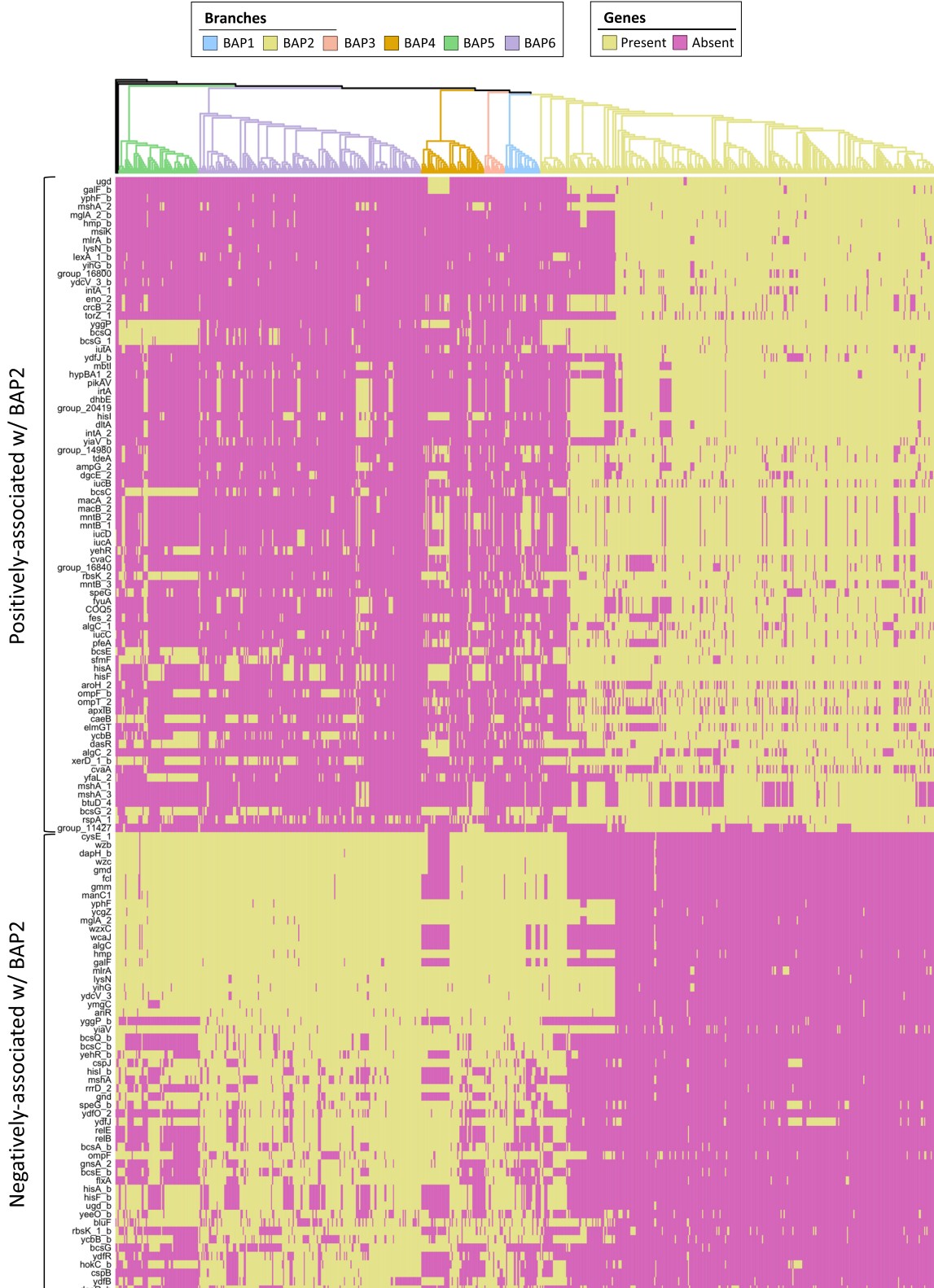

**Fig. 6 Genes associated with the BAP2 cluster.** Presence/absence (yellow/magenta) of genes positively and negatively associated with the BAP2 cluster mapped to the phylogeny. Tree branches indicate BAP clusters.

compared human source sequences (ExPEC, other) with pairwise SNP counts of 20 or less to sequences from the remaining 13 source categories. For perspective, 20 SNPs across the 2.8Mbp core gene alignment constitutes a core nucleotide divergence of just 0.0007%. We identified 135 pairwise cases of ≤20 SNPs that

occurred between 26 humans (15 ExPEC, 11 other) sequences and 34 non-human sequences representing 11 of the 13 non-human sources (Fig. 9). This analysis revealed a close linkage between geographically and temporally distinct strains. One cluster comprised 8 ExPEC from Denmark and 3 human others

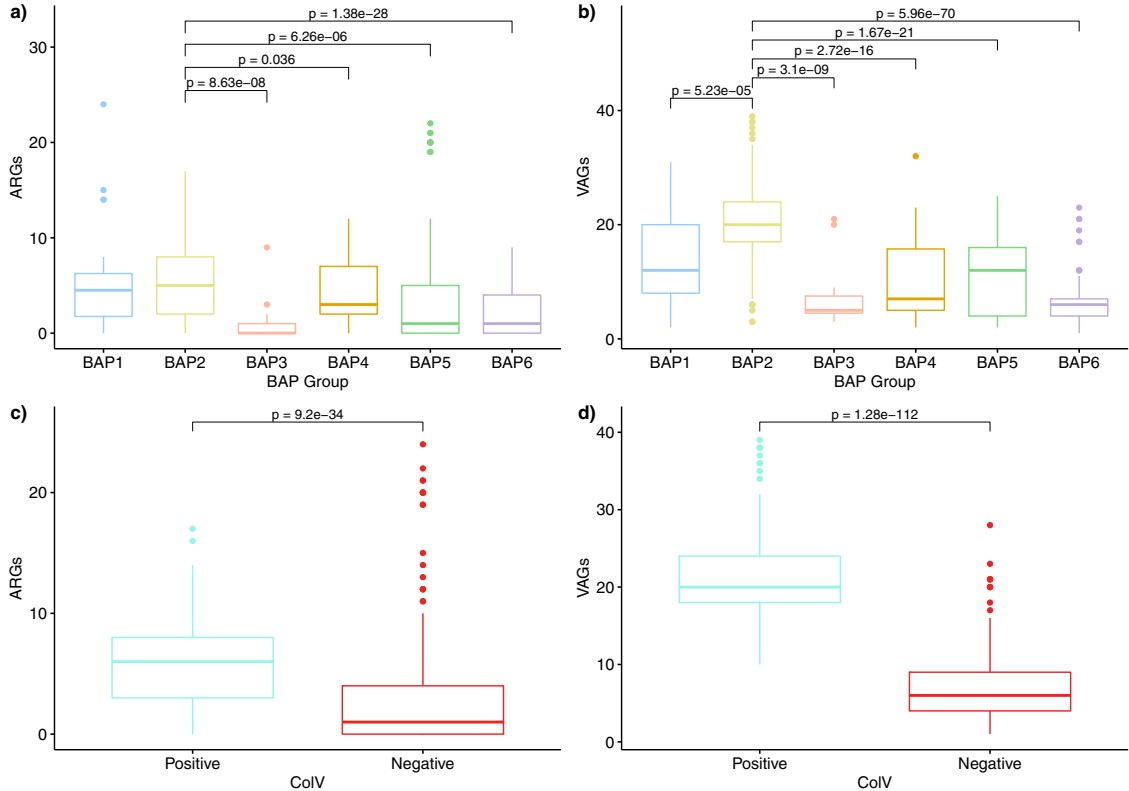

**Fig. 7 Relationships of ARG and VAG carriage with BAP cluster and ColV carriage.** Boxplots comparing total ARG and VAG counts by BAP clusters (**a, b**; $n = 750$ biologically independent ST58 genome sequences) and ColV status for (**c, d**; $n = 752$ biologically independent ST58 genome sequences). **a** ARG carriage by BAP cluster (Pairwise Wilcoxon test with Benjamini–Hochberg adjusted $p$-values); **b** VAG carriage by BAP cluster; (Pairwise Wilcoxon test with Benjamini–Hochberg adjusted $p$-values); **c** ARG carriage by ColV carriage (Two-sided Wilcoxon Rank-sum test); **d** VAG carriage by ColV carriage (Two-sided Wilcoxon Rank-sum test). Upper and lower limits of the boxes represent 75th and 25th quartile, centre line represents the median, whiskers extend to 1.5 × IQR and values outside these ranges are represented by dots. ARG maximum value is 24 and minimum value is 0. VAG maximum value is 39 and minimum value is 1. *P*-values for significant differences are shown. (n.b. Only significant differences between BAP2 and other groups are annotated with *p*-values in **a** and **b**.

(2 faecal swabs, 1 sputum; patient health status unknown) from Thailand with close linkages to poultry and porcine strains from the United States as well as with water and bovine origin strains from Sweden. Among these, ExPEC strain ESBL20140051 from Denmark was separated by only a single SNP from three water strains and one bovine strain, all of which were from Sweden. This is particularly striking given the low numbers of human-origin sequences in the collection and implies that far more genomic links exist between ST58 that cause infections and those present in animal and environmental reservoirs.

## Discussion

Here, in a genomic epidemiological study, we examined 752 whole genomes of *E. coli* ST58 from a variety of sources in order to provide some explanation for its emergence as a human pathogen. We identified the BAP2 cluster; a large, divergent lineage of strains with a broad host range. This cluster had near-ubiquitous carriage of ColV-like plasmids, reduced diversity of adaptive colonisation-related traits such as *fimH* and serotypes, and distinctive accessory gene content, including carriage of yersiniabactin genomic island marker genes and an additional prophage. Among the collection's strains, the BAP2 cluster contained the vast majority of strains isolated from extra-intestinal infections, poultry and swine. Our data indicate that complex interactions between mobile genetic elements, phylogenomic background, and various hosts comprise the mechanisms and

networks through which *E. coli* ST58 has emerged as a human pathogen[56].

**Limitations.** Epidemiological studies that leverage publicly available genomic data are typically beset by uncertainty as to whether the total dataset represents a reliable proxy for the genomic and source diversity of the entire population. Whilst our source distribution was dominated by cattle, poultry and swine, we believe we have observed most of the genomic diversity present in ST58 because of the large pangenome, variety of ARGs, VAGs and plasmid replicons, and clear population structure. The major question remaining is whether most ST58 from human faeces and ExPEC belong to the BAP2 cluster. Though not included in our study collection, ST58 sepsis isolates from multiple hospitals in Paris predominantly display O8/O9:H25 serotypes, as well as ColV and HPI marker genes, all of which were typical of BAP2 ST58[3]. In conjunction with our data, this tends to suggest that most ST58 ExPEC will belong to this cluster, however, an expanded dataset with more commensal and pathogenic human sequences would increase our confidence. Some form of virulence characterisation of non-clinical isolates in the BAP2 cluster would have been desirable in order to solidify the linkage between their genomic traits and phenotypes; however, this was outside the scope of the study. Nonetheless, the sharing of genomic elements with well-documented virulence traits between clinical and non-clinical isolates in the cluster strongly supports a degree of innate virulence. Similarly, though

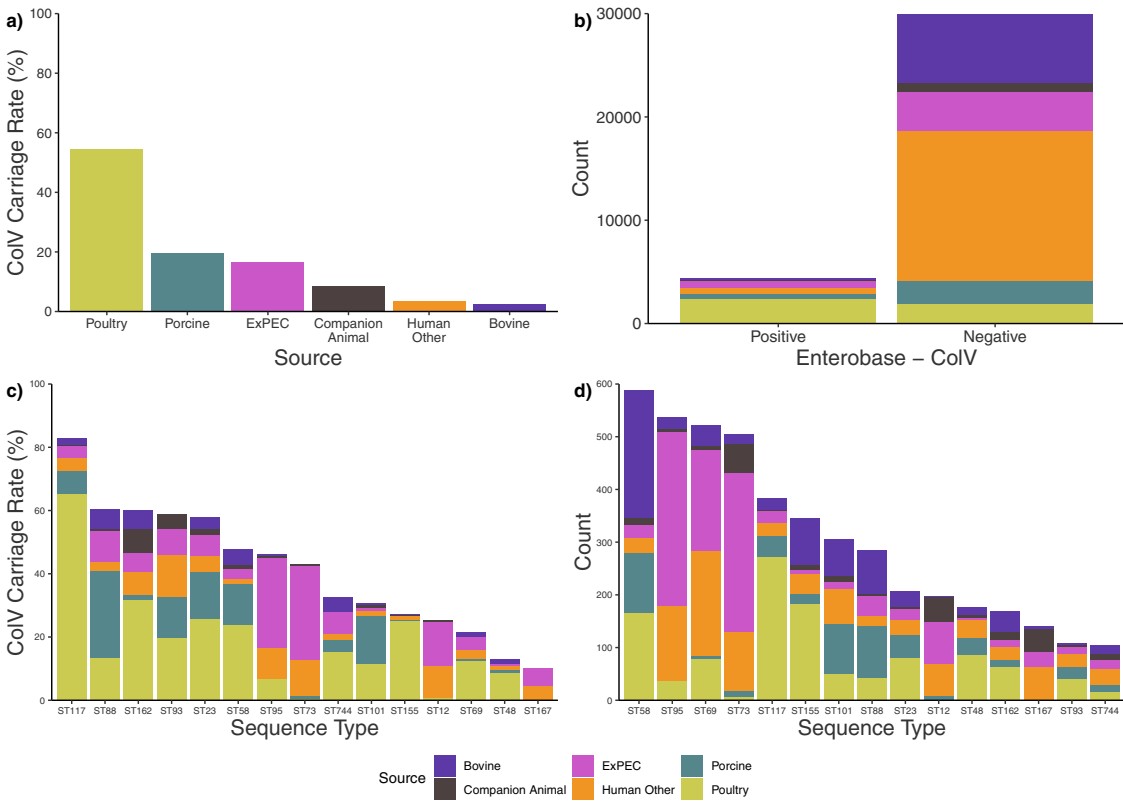

**Fig. 8 ColV carriage in 34,364 *E. coli* genomes.** Summary of sources, STs and ColV carriage from 34,364 Enterobase *E. coli* assemblies. **a** Proportional ColV carriage in each source; **b** Absolute ColV carriage stratified by source; **c** Proportional ColV carriage in the most abundant ColV carrying STs, stratified by source; **d** Absolute counts of STs from **c**, stratified by source.

antimicrobial resistance phenotypes were not available for all strains, concordance between acquired ARG presence and resistance phenotype is typically very high[57,58].

**ColV plasmids in ST58.** Our phylogenetic analysis revealed that the BAP2 cluster of ST58 strains has acquired a diversity of ColV plasmids. The majority of the ColV plasmids observed carried the full repertoire of archetypal ColV genes and operons. The importance of ColV and plasmids generally in the evolution of multiple pandemic ExPEC lineages is increasingly acknowledged[49,59,60]. Carriage of ColV plasmids may prime strains to cause extra-intestinal infections in humans. ColV plasmids typically encode multiple genetic loci including siderophores involved in iron acquisition and transport (aerobactin and salmochelin), *iss* conferring copy number-associated increased survival in human serum, outer membrane vesicle and protease production (*hlyF* and *ompT*) and putative ABC transporter *ets*[48]. This gene repertoire allows strains that acquire them to develop increased virulence in animal models of neonatal meningitis, urinary tract infection and sepsis[53,61], increased killing of chicken embryos, increased growth in urine and colonise the murine kidney[51]. Aerobactin specifically is associated with cystitis, pyelonephritis and bacteraemia[62]. Carriage of yersiniabactin (*fyuA, irp2*, HPI) by the vast majority of strains in the BAP2 cluster also strongly supports their innate virulence[46]. Furthermore, the ColV-based *sit* operon is associated with innate virulence[46]. In addition to pathogenic properties, ColV+ *E. coli* have been shown to outcompete ColV- strains in the human gut[63]. Aerobactin and salmochelin carriage are likely to play a role here, being implicated in intestinal persistence and gut colonisation[64–66]. In this regard, ColV carriage may not only

contribute to BAP2 ST58 pathogenicity but also to lineage expansion via the presence of factors advantageous for intestinal colonisation and persistence.

**Implications for antimicrobial resistance.** Our findings, together with the literature, suggest the acquisition of ColV plasmids heightens the likelihood that a strain will also acquire AMR determinants. In our ST58 genome collection, we observed that, on average, ColV+ strains carried more ARGs than ColV− strains. In line with this, ARG loci primed for further resistance gene acquisition (e.g. co-carriage of *intI1* (class 1 integron), *dfrA5* (a trimethoprim resistance gene) and a globally disseminated IS*26* deletion signature[67]) could be localised to the same assembly contig as ColV genes and were observed in nearly a third of our ColV+ sequences. ARG loci such as these on plasmids tend to act as hotspots for stepwise gene acquisition via multiple mechanisms[68–70]. In healthy humans, ARG loci evolving in conjunction with diverse mobile genetic elements (MGEs), such as IS*26*, on different lineages of ColV plasmids have been described in commensal strains of *E. coli* isolated from the faeces of healthy humans[48,71,72]. ColV plasmids also circulate in the environment with determinants conferring resistance to critically important antimicrobials—observed in an MDR ST58 strain isolated from a pig[73] (ColV-like plasmid carrying $bla_{\text{CTX-M-15}}$) and an MDR *E. coli* ST131-*H*22 strain isolated from agricultural soil[74] (a tranposable IS*Ecp1*-$bla_{\text{CTX-M-15}}$ unit inserted upstream of an IS*26*-truncated copy of *Tn2* on an F2:B1 ColV plasmid backbone). Overall this highlights the risk of ColV plasmids acting as backbones for the acquisition and dissemination of ARGs, though they are most likely selected for traits other than AMR in the first instance.

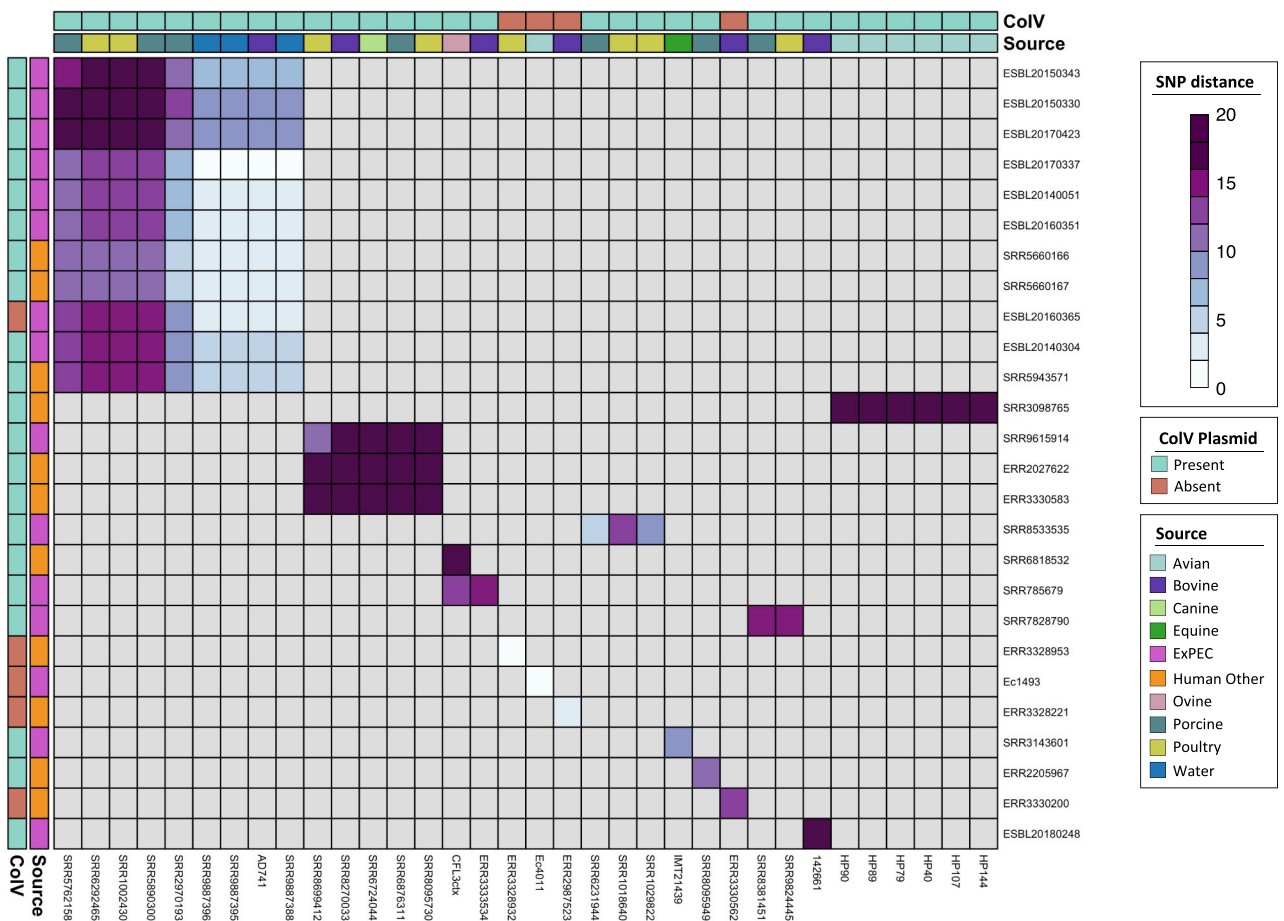

**Fig. 9 Close relationships between human and non-human ST58.** Heatmap of human vs. non-human strain pairs differing by ≤20 SNPs. Scale from white to dark purple represents SNPs from lowest to highest. Grey squares represent more distantly related sequence pairs. Heatmap is clustered to show groups of closely related sequences.

Our analysis of ST58 highlights the importance of screening for AMR in ExPEC without selective bias. Many studies select isolates for study on the basis of their carrying ESBL-encoding genes. However, such methodology can overlook precursor conditions that allow for resistance to critically important antimicrobials to be acquired. This is demonstrated above by ColV carriage as well as IncI plasmids carried by ST58. IncI1 plasmids are known primarily for their carriage of ESBL genes ($bla_{CTX-M-1}$, $bla_{CTX-M-15}$ and $bla_{CMY-2}$). The extent of reporting on ESBL-carrying IncI1 plasmids in ST58 would suggest these plasmids are common in ST58[8,13,35,39]. Interestingly, we found that whilst IncI1 plasmids are found in 30% of ST58, no ESBL gene was present in greater than 8% of sequences. This suggests that IncI1 plasmids were selected in ST58 in sources where ESBL-selection pressure was not a factor, yet these plasmids may provide a convenient scaffold for ESBL gene acquisition when the relevant selection pressure is encountered. This is conceptually reflective of the evolution of ARG loci on some ColV plasmids, which evidently occurred subsequent to their primary evolution and widespread dissemination[48,66]. Thus, understanding the ecology and lineage associations of plasmids without antimicrobial selection bias is critical to a holistic understanding of factors that influence the emergence and spread of ARGs.

**Ecology of ColV plasmids**. Our findings illustrate the relationship between the source, MGEs (mediators of horizontal gene transfer) and genomic background in pathogen emergence. The Enterobase analysis supports the role of ColV in pathogenicity in a proportion of human extra-intestinal infections. Of 34,364 draft *E. coli* genome assemblies from Enterobase, ColV plasmids were present in 16% of ExPEC strains. In ST58, we observed the presence of ColV plasmids in more than 75% of the ExPEC strains. However, ExPEC strains are not the dominant reservoirs of ColV plasmids. Rather, our data supports the widely held opinion that poultry and associated meat are the major reservoirs for ColV plasmids with a carriage rate of 55%[54,75]. Unexpectedly, we found that porcine sources were also a non-trivial secondary reservoir of ColV plasmids, with 20% carriage. ColV plasmids have previously been reported in pigs; however, it is unclear whether ColV plasmids were historically common in production systems or whether they represent a relatively recent, direct or indirect incursion from poultry[76,77]. Further analysis of plasmid diversity in the species collection could be useful in this regard. Consistent with the Enterobase analysis, our collection of 752 genomes of ST58 revealed that poultry and porcine-sourced ST58 were more often ColV+ than ColV-. Interaction between ColV plasmids, phylogenomic background and source was also evident in the variable proportions of isolation sources displayed by the major ColV+ STs.

To what extent is the human gut a reservoir of ColV+ ST58 and ColV plasmids in general? It remains difficult to quantify. Human faecal commensals are the origin of the vast majority of ExPEC infections[78,79] and yet very few sequences are available or

identifiable online due to the strong bias towards human diarrhoeagenic isolates and lack of metadata, respectively. Nonetheless, evidence pointing to the human gut as a reservoir comes from our study and others[48,63]. As previously mentioned, the carriage of multiple siderophore systems such as aerobactin and salmochelin is considered to be advantageous for gut colonisation[64–66]. Detailed information on ColV carriage in large collections of geographically diverse human faecal commensals is urgently required as it is the critical missing link between primary sources of ColV plasmids and ColV+ ExPEC infections.

**Plasmids, sources and phylogeny in pathogen emergence.** The importance of interrelation between plasmids and sources for understanding pathogen emergence is highlighted by recent literature. Evolutionary models demonstrate the role of horizontal gene transfer and microbial migration in decoupling ecologically important traits from genomic backgrounds in a specific niche and allowing horizontal sweeps of these traits under selective pressure[80]. Experimental work similarly demonstrated that different microbial habitats carry distinct sets of MGEs, which could drive the evolution of immigrant *E. coli* by providing a pool of MGE-associated functional elements advantageous to survival in that niche[81]. Shaw and Matlock have also reported the effect of niche on the distribution of Enterobacterial plasmids and F plasmids, respectively[82,83].

The dominance of poultry and porcine sources in the ST58 BAP2 cluster is therefore likely to be because ColV plasmids and other BAP2-associated genes are abundant and provide a fitness advantage within niches associated with these hosts. The lower diversity of key adaptive traits (e.g. *fimH* alleles and serotypes) in the BAP2 cluster compared to other clusters is suggestive of evolutionary convergence commensurate with selective pressure exerted by a specific host or hosts[84,85]. Overall this suggests that poultry and swine have influenced the evolution and emergence of BAP2 ST58.

As well as source, the phylogenomic background is also known to affect the propensity for certain lineages to horizontally acquire genetic material and conversely horizontal acquisition influences core genomic changes[86]. Phylogroup B1, to which ST58 belongs, is considered to be particularly receptive to horizontally acquired genetic material[81]. The abundance of ColV plasmids in one particular lineage however suggests that BAP2 ST58 may be more susceptible to plasmid acquisition, more suited to niches where ColV plasmids are abundant or a combination of both. Similarly, the phylogenetic divergence of the cluster may have been influenced by compensatory SNPs accumulated as a result of the acquisition of ColV plasmids and other MGEs[86].

With all of the above in mind, it appears that the ST58 BAP2 cluster has emerged via multiple horizontal acquisitions of ColV plasmids, phage and additional accessory genes as well as core genomic adaptations both favourable to and resulting from these evolutionary events. We strongly suspect that lineage expansion has been significantly influenced by ColV plasmids and their association with hosts such as poultry and swine. Siderophore iron acquisition systems found on ColV plasmids and the HPI are also likely to have been influential via their conferral of both fitness-related and pathogenic traits. The order in which specific genomic events actually occur in conjunction with transfer between different sources is unknown, but can be conceptualised as a constant push-pull relationship between core genomic adaptations and HGT occurring within a complex network of interacting sources and MGEs under selection within and between them.

## Conclusion

Our epidemiological data indicate that *E. coli* ST58 has emerged as a prominent sequence type and human pathogen through the interplay of mobile genetic elements, ecology, and genomic background. We identified a sub-lineage of ST58 that has acquired extrachromosomal and chromosomally-located mobile genetic elements, most notably a diversity of ColV plasmids and the Yersiniabactin High Pathonenicity Island. Poultry and swine are implicated in the emergence of this distinctive sub-lineage, with the contribution of the collective human gut yet to be fully appreciated. Carriage of ColV plasmids in ST58 exemplify selection offering *E. coli* fitness advantages where it is commensal or free-living and virulence capabilities once in the urinary tract or blood. Although ColV plasmids may not be primarily selected for their capacity to confer resistance to antimicrobials, their frequent carriage of ARGs is an additional threat to those who develop infections caused by ColV+ ST58. Our study highlights the importance of genomic epidemiological investigations that avoid antimicrobial selection bias and take plasmid ecology and non-human sources into consideration in order to develop a holistic understanding of factors influencing *E. coli* pathogenesis.

## Methods

**Collaborator sequences.** A global collection of 190 ST58 *E. coli* sequences was compiled from in-house datasets and collaborators in Australia, Canada, Czech Republic, Denmark, France, Germany, Netherlands, Poland, Romania, Sweden, United Kingdom and United States. These sequences represented a wide range of sources including human urinary tract and blood infections, wild birds, cattle, companion animals, produce, swine, poultry, wastewater and surface water. Detailed methods regarding culture and DNA isolation for the generation of in-house sequences are available in their respective publications listed as PMID accessions in Supplementary Data 1. All data from collaborators were sequenced on Illumina platforms (see SRA entries for specific instruments) and was received as raw *fastq* paired reads. Individual strain and DNA isolation methods were not provided for collaborator sequences. In-house and collaborator reads were directly uploaded to an Enterobase workspace for quality control (QC) and preliminary analyses in the context of other ST58 sequences. Twelve sequences were excluded as they were determined not to belong to ST58, failed assembly or QC performed by Enterobase. Eight sequences were excluded in preliminary phylogenetic analyses. The final number of 'in-house' sequences in the collection was 178. Of these, 158 were previously unpublished and have been uploaded to SRA under BioProject PRJNA727368. Full metadata and accession numbers for all sequences are available in Supplementary Data 1.

**Publicly available sequences.** Enterobase was queried on 04/12/2019 for released ST58 whole-genome sequences with available metadata for source, collection year, continent and country. SRA and ENA accession numbers were extracted and parallel-fastq-dump 0.6.6 was used to download 614 read sets with the following flags:–skip-technical,–read-filter pass,–dumpbase,–split-files,–clip. Enterobase sequences were named for analysis with their NCBI SRA or EBI ENA accession number beginning with SRR or ERR, respectively. Full metadata and accession numbers for these sequences are available in Supplementary Data 1.

**Curation and metadata processing.** To create consistency and ensure cogency within source information from Enterobase we curated the metadata in R 3.6.3 with a range of regular expression substitutions and some unavoidable manual curation utilising the three source information columns from Enterobase to classify sequences by niche and source. Defined niches and their respective sources in parentheses were: livestock (bovine, equine, porcine, poultry, livestock other, ovine); human (ExPEC, human other); wild animal (avian, other mammal); companion animal (canine, feline, avian); food (plant); environment (water, wastewater). Forty sequences were excluded during this process due to irreconcilable source information. The scripts used to process the raw metadata, as well as the pre- and post-manual curation datasets have been made available as a GitHub repository for reproducibility and transparency (See 'Data availability' and 'Code availability' below). The final collection numbered 752 sequences comprising 178 in-house/collaborator sequences and 574 publicly available sequences.

**De novo assembly and annotation.** Raw sequence reads were de novo assembled with Shovill 1.0.4 with default settings and a minimum contig length of 200 bp. The resulting assemblies were annotated by Prokka 1.14.5, run with default settings and the–kingdom and–genus flags set to 'Bacteria' and 'Escherichia', respectively.

**Core and pangenome identification.** The collection of 752 prokka-annotated sequences and outgroup strain MOD-EC5019 (ST155) were analysed with Roary 3.13.0 to infer core and pangenomes of *E. coli* ST58[87]. -v and -e flags were used to produce an alignment of core genes with MAFFT 7.455, which formed the basis of

phylogenomic and SNP analyses described below. Aside from these flags, default settings were used with paralog splitting on, 95% minimum identity for BLASTp and core genes defined as those present in 99% of isolates (this prevents the outgroup strain from affecting core genome estimation). The resulting core gene alignment comprised 2,807,790 bp and 3023 core genes.

**Phylogenomic and SNP analyses**. A maximum-likelihood phylogenetic tree was inferred from core gene alignment with IQTree 2.0.3 using the GTR + F + R4 model of nucleotide substitution and 1000 bootstrap replicates and rooted on the outgroup strain. fastbaps 1.0.6 was utilised to define clusters from the core gene alignment conditioned on the core gene tree and these designations were subsequently used in downstream analyses[88]. SNPs were identified by running snp-sites 2.5.1 on the core gene alignment, generating a core SNP alignment of 30,771 SNP sites. Pairwise SNPs between strains were counted from the core SNP alignment with snp-dists 0.6.3.

**Pan-GWAS**. Scoary calculates associations between gene presence/absence (as determined by Roary) and phenotypic or metadata traits of the strains. We defined the fastbaps clusters as traits and used Scoary 1.6.16 with the –no_pairwise flag (to identify over-representation as opposed to causation) and a Benjamini–Hochberg corrected p-value threshold of 1E-50 to identify genes that were over- or underrepresented in those clusters.

**Gene and mutation screening**. Antimicrobial resistance genes, virulence-associated genes, plasmid replicons, F plasmid types and serotypes were identified via a read-mapping approach with ARIBA 2.13.3[89], using publicly available ResFinder, VirulenceFinder, PlasmidFinder and SerotypeFinder databases from the Centre for Genomic Epidemiology and custom databases available at https://github.com/maxlcummins/custom_DBs/blob/master/EC_custom.fa[90–94]. SNP-mediated resistance genotypes were predicted with PointFinder 3.1.0[95], downloaded as a Python script from https://bitbucket.org/genomicepidemiology/pointfinder/src/master/. *fimH* alleles were identified by Enterobase. ABRicate 0.9.8, a BLAST-based screening tool, was additionally used with default settings to identify IncI1 pMLST and the *dfrA5*-IS26 deletion signature.

**Inference of ColV plasmid carriage**. We used ABRicate 0.9.8 with default settings to implement the ColV screening methodology described by Liu *et al* to identify de novo assemblies that putatively carried a ColV plasmid[75]. Briefly, the Liu criteria considers a strain ColV-positive if it carries at least one or more genes from four or more of the following six gene sets (i) *cvaABC* and *cvi* (the ColV operon), (ii) *iroBCDEN* (the salmochelin operon), (iii) *iucABCD* and *iutA* (the aerobactin operon), (iv) *etsABC*, (v) *ompT* and *hlyF*, and (vi) *sitABCD*. Thresholds of ≥ 90% nucleotide identity and ≥95% length coverage were applied post-ABRicate in RStudio 1.4.1106 with R 4.0.5 to determine positive hits for a gene and sum the group counts to infer presence or absence.

To strengthen the evidence for ColV plasmids in our collection we aligned assemblies to the backbone of pCERC4, an archetypal F2:B1 ColV plasmid isolated from an ST95 *E. coli* strain from healthy human faecal flora. Briefly, the 126,270 bp pCERC4 backbone was defined by removing resistance regions from the complete pCERC4 sequence (RefSeq: NZ_KU578032.1) as described by Moran and Hall[48]. Assemblies were aligned against the backbone with ABRicate and outputs concatenated for import into R. We used a custom R script to extract hit ranges for each sequence and represented them as a series of percentage nucleotide identities across 100 bp segments of the plasmid backbone. These percentages were then visualised against a schematic backbone of pCERC4, generated with SnapGene 5.2 (GSL Biotech), as a heatmap in ggtree.

**Enterobase ColV analysis**. To compare ColV carriage across *E. coli* STs we filtered the Enterobase *Escherichia/Shigella* database for strains deposited prior to January 12, 2020. To be included in the study, strains were required to have metadata relating to their source, geographical and temporal origins. Serotypes, Achtman MLST and *fimH* alleles were also downloaded from Enterobase. Sources were curated in R as well as manually in Excel in order to reduce the heterogenous Enterobase metadata into six categories: Bovine, Porcine, Poultry, ExPEC, Human Other and Companion Animal. Following removal of *Shigella sp.,* 35,254 genomic assemblies were then downloaded from Enterobase using a third-party custom script available at https://github.com/C-Connor/EnterobaseGenomeAssemblyDownload. Genomes less than 4.5 Mbp or greater than 6.5 Mbp were excluded. ABRicate 0.9.8 was used to screen for ColV related genes and the Liu criteria was applied to infer ColV carriage as described above. Metadata, accession numbers and additional information on the cohort of strains under analysis are available in Supplementary Data 4.

**Statistics**. All statistical analysis were performed in R 4.0.5 and scripts are available as described in 'Code availability' below. The Kruskal-Wallis test was used to determine whether the mean of total ARGs or VAGs per sequence was different based on BAP group membership and between ColV+/− status. Pairwise

Wilcoxon test with Benjamini–Hochberg p-value correction for multiple testing was then used to calculate the significance of the pairwise differences between BAP groups. Two-sided Wilcoxon rank-sum test was used to calculate the significance of the difference between ColV+ and ColV− sequences.

**Data processing and visualisation**. All processing, analyses and data visualisation were performed in RStudio 1.4.1106 with R 4.0.5 and can be reproduced with the scripts linked in 'Code availability'. Some manual editing of figures for consistency of presentation was performed in Microsoft PowerPoint.

**Reporting summary**. Further information on research design is available in the Nature Research Reporting Summary linked to this article.

## Data availability

All ST58 raw sequence read data generated for the first time in this study have been deposited in NCBI BioProject PRJNA727368 with individual accession numbers available in Supplementary Data 1. All other ST58 raw sequence read data used in this study are available via either the NCBI Sequence Read Archive [https://www.ncbi.nlm.nih.gov/sra] or the EMBL-EBI European Nucleotide Archive [https://www.ebi.ac.uk/ena/browser/home] via accession numbers listed in Supplementary Data 1. Genome assemblies from the Enterobase collection are available at https://enterobase.warwick.ac.uk/species/index/ecoli via assembly barcodes listed in Supplementary Data 4. Source data are provided with this paper.

## Code availability

To support the reproducibility of this work we have made all the raw data generated from the original sequence reads, and the R scripts used to process, analyse and visualise this data available at https://github.com/CJREID/ST58_project.

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

## Acknowledgements

We would like to thank: Frank Hansen, Karin Sixhøj Pedersen and Hülya Kaya for their excellent technical assistance; Jonas Bonnedahl, Linköping University; Sara Byfors, Public Health Agency of Sweden; Catarina Flink and Mia Egervärn, Swedish Food Agency. Ivan Literak, Jaroslav Hrabak, Iva Kutilova and Ivana Jamborova for providing strains and help in the laboratory. Kay Anantanawat for genome sequencing support at iThree Institute. Ian Charles and Garry Myers for their helpful comments on the manuscript. Fiona MacIver for editing. This study was partially funded by Czech Science Foundation Grant no. 18-23532 S awarded to M.D. It was also supported by an Australian Government Medical Research Future Fund project (MRFF75873), the Australian Centre for Genomic Epidemiology Microbiology (AusGEM), a strategic research initiative between the New South Wales Department of Primary Industries and the University of Technology Sydney and Australian Research Council Linkage Project LP150100912.

## Author contributions

C.J.R. contributed conceptualisation, data curation, formal analysis, investigation methodology, project administration, software, validation, visualisation, writing—original draft and writing review and editing; M.L.C. contributed data curation, investigation, methodology, software, validation, writing—review and editing; S.B. contributed data curation, investigation, resources, writing—review and editing; M.B. contributed data curation, investigation, resources, writing—review and editing; H.H. contributed data curation, investigation, resources, writing—review and editing; A.M.H. contributed data curation, investigation, resources; L.R. contributed data curation, investigation, resources; S.H. contributed data curation, investigation, resources, writing—review and editing; T.B. contributed data curation, investigation, resources, writing—review and editing; K.N. contributed data curation, investigation; M.H. contributed data curation, investigation, resources, writing—review and editing; J.-Y.M. contributed data curation, investigation, resources; A.B. contributed data curation, investigation; G.B.M. contributed data curation, investigation; A.-K.S. contributed data curation, investigation; S.S. contributed contributed conceptualisation, data curation, project administration, resources, supervision, writing—review and editing; M.D. contributed conceptualisation, data curation, funding acquisition, project administration, resources, supervision, writing—review and editing; S.P.D. contributed conceptualisation, funding acquisition, project administration, resources, supervision, writing—review and editing.

## Competing interests

The authors declare no competing interests.
