## [Peer Review File · Nature Communications]

A role for ColV plasmids in the evolution of pathogenic *Escherichia coli* ST58REVIEWER COMMENTS

Reviewer #1 (Remarks to the Author):

THE SUMMARY:

The authors present a complex manuscript arguing that "ColV plasmids drive the evolution of a pathogenic lineage of Escherichia coli B1-ST58"

The manuscript describes a playbook of a particular E. coli that undergo a set of genomic changes that converts from what would probably be an enteric commensal into a pathogenic enterintestinal. This describing the mechanisms of this transition would be of interest to the field.

First of all, the methods are technically correct and I applaud the authors. Steps of genome assembly, comparative genomes and phylogenetics are done well; barring some issues below (See "SPECIFIC COMMENTS"). My issue with this manuscript is that the interpretation of these results tends to be overstated, and that in turn affects the implications of the study (See "INTERPRETATION").

SPECIFIC COMMENTS:

I should note that there is no adequate description of how the in-house data set was collected. How was the genome sequencing performed for in house "datasets"? I.e no methods section for culturing, DNA extraction or library preparation. Methods sections implies these are unpublished data so this should be explained. If there multiple providers of cultures > sequencing, then there will need to be separate sections for each provider. Maybe the sequencing data (i.e. FASTQ) was just provided to the authors so they don't know. This is fine, but it needs to be explained in some way. What was the sample collection strategy, even if this data has been frankensteined from other previous projects. This has to be dealt with in the text. As it is currently written, the genomic data seems to appear out of thin air. This will not do.

I am perplexed why the authors mention that 18 unique genes are introduced by each genome (and include plots to demonstrate an open pangenome) but then do not incorporate that into any larger insight or discussion. The manuscript is large enough as it is, perhaps this can be removed?

And the Figure 2 legend needs more details i.e. maximum-likelihood phylogeny, how many tips, etc.

As a final note, Please scale your figures, even the supplementary ones submitted separately, as A4 always. It is impossible, as a reviewer, to refer between the main text and the separate figures because one is the size of a page and the other is the size of a billboard.

INTERPRETATION:

The study is based on presumption that acquiring a ColV plasmid means you have a more pathogenic E. coli. And ColV acquisition is accompanied by some known interesting genes (iron acquisition). My first issue is with the idea of acquisition. All the trees presented here do not seem to be rooted to a suitable outgroup, so say Figure 2, it shows a clade at the base (which has ColV) then a number of clades (BAPS clusters) without, then this large BAPS3 with the plasmid again. If the true root of the tree was closer to the BAPS3 cluster than the topology would be reversed, so it appears that having ColV is an ancestral trait and then the plasmid is lost several times. Given the number of times the plasmid is coming and going, both scenarios are equally parsimonious. It is Critical that the authors establish which part of the tree is basal (good outgroup, proper rooting, maybe use of time trees?) because it is only then that they can assert directionality -- that something acquired something. This weakens statements that cattle would be the reservoir.

Yes, I do concede that in the literature that ColV is associated with human disease, and the authors also want to promote that fact. But if anything their data contradicts it (see figure 8) isolates with ColV across STs seem to be in a number of different hosts, and the proportion of Human EXPEC/Other varies from ST to ST. I would expect if ColV in of itself was really causative, that pink EXPEC bar should be the majority in almost every ST surveyed. On the other hand, The authors state that bovines are the primary host of ST58 and in that niche there is a mix of ColV positive and negative isolates. Let's try this another way. Figure 2 presents BAPS4 cluster as

having some proportion of ColV positive strains, yet it doesn't seem to be particularly different from the other clusters in any of the figures presented. Perhaps this might be done to the use of the arbitrary definitions of the BAPS clusters. I would be curious if there is a difference between COLV+ve in BAPS4 vs COLV negative. Since they are so closely related.

I know the literature supports ColV plasmid acquisition as then conferring some pathogenic potential. But the genomic data presented here just doesn't give enough information to say if this is the case. The authors use statements "BAPS3 sequences on average carry significantly more ARGs than other BAP clusters ... " but then do not quantify significance. They state that 16% of ExPEC in their survey have ColV plasmids -- is that significant in some way? No statistical test is provided. No laboratory experiment is provided. No framework for testing this hypothesis is provided. What would be the likelihood of this correlation being observed by random chance?

To me, if anything, it appears that IF the BAPS3 clade is truly special (and that is an IF) then it could be some other set of genomic changes and the ColV plasmid is picked up by-the-by, a proxy marker to something happening elsewhere.

The final sentences of the conclusions regarding the implications of the work are not supported by the results. For instance, detecting ARGs in particular E. coli does not mean that that resistance will lead to increase in clinical cases and be "Deleterious to human health" and there are not a large number of human associated ST 58 so it does not seem to be an emerging pathogen.

Reviewer #2 (Remarks to the Author):

The paper of Reid et al. reports a phylogenomic analysis of a panel of 752 E. coli B1-ST58 strains, a clone emerging in animal and human pathologies while becoming resistant to antibiotics. The topic is of interest, although not new. The work is well done and the results are discussed fairly. I have however two major concerns.

First, the ST58 according to the Warwick scheme corresponds to the ST24 and 87 according to the Pasteur Institute scheme. Together with the ST155 (Warwick nomenclature), they constitute the CC87 (Pasteur nomenclature). The emergence of this clonal complex has been first published in 2015 (PMID: 26613786) with several phenotypic tests including conjugation assays that should be discussed. Its recent increase in human bloodstream infections is clearly noted in a 12-year period (PMID: 33952335). There is also an important literature by the Bonacorsi group on the pS88/pS88-like plasmid that should be discussed (see for example PMID: 24086343). In sum, please position your work according to the global knowledge.

Second, I am not convinced by the title and more precisely that it is the ColV plasmids that drive the evolution of the lineage. Pathogenicity in E. coli (excepting perhaps Shigella) is a multifactorial phenomenon. I think numerous other genes (as those of the HPI) are probably involved in the process.

Erick Denamur

Reviewer #3 (Remarks to the Author):

Reid et al describes the detection and characterisation of ColV plasmids in E. coli ST58 from global

collection of isolates. The genomic analysis of the isolates are extensive and comprehensive. However, there are some concerns regarding interpretation, relevance, novelty of the findings and number of conclusions that cannot be substantiated with the current study design and epidemiological data.

- The study is largely observational with large number of ST58 isolates.
- The major finding from this manuscript is that large number of ST58 carry ColV plasmid. This has been already reported by the authors in a number of recent publications (McKinnon et al 2018 Int J Antimicrob Agents 52(3):430-435; Wyrsh et al Microbial Genomics 2020;6; Reid et al 2020 Front. Microbiol;)
- There is no experimental in-vivo study undertaken to clarify the virulence of these strain/ plasmid using murine model. Which is one of the common approaches taken to clarify virulence of ExPEC isolates
- ColV has been well characterised (Johnson TJ J Bacteriology. 188, 2) as a potential virulence plasmid and ST58 has been previously reported as a potential animal pathogen and it is not surprising that the strain carry ColV plasmid. (Fuentes-Castillo et al Transboundary and Emerging Disease).
- Coverage biases in Illumina sequencing and its impact on quality of data and inferences are not addressed
- A number of isolates carried ESBL genes, could the over representation of ST58 in the study could be attributed to antimicrobial resistance rather than virulence of this isolates. This has been the case for uropathogenic E. coli ST131, a moderately virulent strain becoming a pandemic ExPEC strain due to the carriage of fluoroquinolone and/third generation cephalosporin resistance?
- What is the fluoroquinolone resistance status of these isolates, that may explain it emerging status? This has also been seen in other emerging ExPECs such as E. coli ST648 (Schaufler et al Antimicrobial Agents and Chemotherapy 2019 63, 6) and ST1193 (Johnson et al 2019 Journal of Clinical Microbiology 57, 5; Johnson et al Antimicrob Agents Chemother 2018 Dec 21;63)
- Could ST58 be a dominant animal associated clone and a sub population of humans may acquire this clone either by direct transfer through the environment/ food as indicated by Borges et al 2019 Foodborne Pathog Dis . 2019 Dec;16(12):813-822.
- ST58 has also been reported to as an atypical EPEC (Jouini et al Antibiotics 2021, 10, 670). How many of the isolates in this study are aEPEC. This need to be clarified.

Specific comments

- Figure 7 is not informative. Carriage of ColV plasmid naturally would increase the putative virulence genes scores. In addition, biases in isolates used in this study (virtue of convenient sampling) could result in difference in carriage of virulence/ antimicrobial resistance genes.
- Lines 415-423 – Virulence of these isolates are not well characterised; carriage of colV alone does not make the strain highly pathogenic. This is evident in the meta analysis where ST58 was a lower ranked ExPEC (ref 2; Manges et al 2019 Clin Microbiol Rev 32)
- Lines 493-507 Isolation from cattle does not mean cattle adapted. The conclusions drawn are weak and not substantiated with the limited data set used in this study
- Lines 510-527. The conclusion on vectors-sources of ColV is not substantiated by this study. There is no experimental evidence from this study or other published literature make this conclusion.
- Lines 442-454. A large number of studies have reported iroN and other siderophore systems

among large number of E. coli strains therefor ColV and iroN unlikely to be associated with grazing resistance

- It is unclear why ColV carriage in humans faecal commensal is urgently needed? ColV has been reported previously and how that information may prevent humans from getting extraintestinal infections is unclear.
- A number of statements are not backed up by references example Lines 415-423
- Lines 542-545. It is unclear how plasmid ecology could assist with risk ranking of bacteria. Perhaps carriage of resistance and virulence assist but just the carriage of plasmids may not help. See Collineau et al Front Microbiol. 2019; 10: 1107 for approaches undertaken for quantitative risk assessment using WGS for food AMR.

Response to reviewer comments

Dear Reviewers,

Thank you kindly for taking the time to review our manuscript and your helpful and challenging comments. The process has been incredibly valuable for us in upgrading and refining the manuscript. We think this is evident in a much stronger revised manuscript.

Reviewer #1 (Remarks to the Author):

THE SUMMARY:

The authors present a complex manuscript arguing that “CoIV plasmids drive the evolution of a pathogenic lineage of Escherichia coli B1-ST58”

The manuscript describes a playbook of a particular E. coli that undergo a set of genomic changes that converts from what would probably be an enteric commensal into a pathogenic extraintestinal. This describing the mechanisms of this transition would be of interest to the field.

First of all, the methods are technically correct and I applaud the authors. Steps of genome assembly, comparative genomes and phylogenetics are done well; barring some issues below (See “SPECIFIC COMMENTS”). My issue with this manuscript is that the interpretation of these results tends to be overstated, and that in turn affects the implications of the study (See “INTERPRETATION”).

SPECIFIC COMMENTS:

Comment: I should note that there is no adequate description of how the in-house data set was collected. How was the genome sequencing performed for in house “datasets”? I.e no methods section for culturing, DNA extraction or library preparation. Methods sections implies these are unpublished data so this should be explained. If there multiple providers of cultures > sequencing, then there will need to be separate sections for each provider. Maybe the sequencing data (i.e. FASTQ) was just provided to the authors so they don't know. This is fine, but it needs to be explained in some way. What was the sample collection strategy, even if this data has been frankensteined from other previous projects. This has to be dealt with in the text. As it is currently written, the genomic data seems to appear out of thin air. This will not do.

Response: We have edited the Methods section to address this. Specifically, we have:

- Described receipt of the raw reads from collaborators
- Added that Enterobase upload was used to verify the quality of the 178 collaborator and in-house datasets
- Added a column in the supplementary datasheet that specifies whether sequences were from In-house, Collaborator or Public
- Added a column in the supplementary datasheet specifying PubMed IDs for in-house sequences that were first described in other publications

Comment: I am perplexed why the authors mention that 18 unique genes are introduced by each genome (and include plots to demonstrate an open pangenome) but then do not incorporate that into any larger insight or discussion. The manuscript is large enough as it is, perhaps this can be removed?

Response: This has now been removed.

Comment: And the Figure 2 legend needs more details i.e. maximum-likelihood phylogeny, how many tips, etc.

Response: Whole figure has been updated in line with other reviewer comments below. Legend now indicates the requested detail i.e maximum likelihood tree, outgroup sequence and number of sequences used.

Comment: As a final note, Please scale your figures, even the supplementary ones submitted separately, as A4 always. It is impossible, as a reviewer, to refer between the main text and the separate figures because one is the size of a page and the other is the size of a billboard.

Response: All figures were exported from R with A4 dimensions 201mm x 270mm but I suspect this issue is because we chose a dpi of 600 which is very high resolution. Have scaled down to 300dpi for resubmission.

INTERPRETATION:

Comment: The study is based on presumption that acquiring a ColV plasmid means you have a more pathogenic E. coli. And ColV acquisition is accompanied by some known interesting genes (iron acquisition). My first issue is with the idea of acquisition. All the trees presented here do not seem to be rooted to a suitable outgroup, so say Figure 2, it shows a clade at the base (which has ColV) then a number of clades (BAPS clusters) without, then this large BAPS3 with the plasmid again. If the true root of the tree was closer to the BAPS3 cluster than the topology would be reversed, so it appears that having ColV is an ancestral trait and then the plasmid is lost several times. Given the number of times the plasmid is coming and going, both scenarios are equally parsimonious. It is Critical that the authors establish which part of the tree is basal (good outgroup, proper rooting, maybe use of time trees?) because it is only then that they can assert directionality -- that something acquired something. This weakens statements that cattle would be the reservoir.

Response: We have re-run the phylogenetic analysis with an ST155 strain as the outgroup. ST155 is a member of the same clonal complex (CC155) as ST58 and so should be closely related enough, yet distinct from the ST58 sequences.

- Outcomes of this analysis include:
 - Highly similar tree topology to the previous trees
 - Division of the phylogeny into 6 major clusters instead of 5

- The previously described BAP3 cluster has been renamed BAP2 and is now larger than before and retains the vast majority of ColV+ strains
- The previous BAP5 cluster is now BAP6 and contains mostly ColV-Bovine-origin sequences as before

Respectfully, we believe there are some misconceptions about ColV plasmids

- ColV does not describe a single plasmid, rather a family of conjugative plasmids with similar gene content that can be differentiated into distinct lineages with independent evolutionary histories via their IncF replicon sequences. Notwithstanding the fact that they are self-transmissible mobile genetic elements by their very nature, the diversity of F plasmid replicons in the BAP2 clade therefore rejects the assertion that carriage could be an ancestral trait as these ColV lineages are present across multiple sequence types of *E. coli* (illustrated in our Enterobase analysis) indicating they are most likely to be acquired elements in almost all cases.
- Furthermore, the presence of ColV plasmids sporadically in other branches of the phylogeny does not negate the near ubiquitous presence of ColV plasmids in the largest evolutionary cluster of ST58. Our findings do not imply that ColV plasmids should be excluded from other lineages of ST58, rather that some ecological and/or genomic changes in the BAP2 cluster have contributed to mass ColV plasmid acquisition.

We do not argue at any point that cattle is ‘the reservoir’ of ST58 - simply that it is the dominant source of isolate found in the BAP6 (formerly BAP5 cluster). Upon inclusion of the requested outgroup strain, this trend remains and is actually slightly more pronounced (69% bovine in that cluster as opposed to 65% prior). Please see also updated Figure 3 showing the obvious differences between BAP2 and BAP6 groups, particularly with respect to ColV carriage.

Comment: Yes, I do concede that in the literature that ColV is associated with human disease, and the authors also want to promote that fact. But if anything their data contradicts it (see figure 8) isolates with ColV across STs seem to be in a number of different hosts, and the proportion of Human EXPEC/Other varies from ST to ST. I would expect if ColV in of itself was really causative, that pink EXPEC bar should be the majority in almost every ST surveyed.

Response: There is a misunderstanding here, in our opinion. The fact that ColV plasmids *contribute* to disease in extra-intestinal sites does not mean that every ST that carries it should be exclusively isolated from cases of extra-intestinal infection. This argument is analogous to, ‘if drunk drivers really cause car crashes then the majority of car crashes should be caused by a drunk driver’, which cannot be true. Each ExPEC infection is unique in the factors that manifest it, but certain factors are known to be influential across multiple infections. We show in figure 8a that ColV is present in ~16% of *E. coli* from extra-intestinal infections, regardless of sequence type and is therefore likely to play a role in those cases but certainly not all cases of ExPEC infection.

At no point in the manuscript do we argue that ColV is the sole cause of extra-intestinal disease. It is well known that all ExPEC are a subset of faecal commensals which cause disease when they enter extra-intestinal sites such as the urinary tract. Their actual virulence is multifactorial and whilst host factors play a primary role, ExPEC also carry a variety of virulence factors which enhance the likelihood that a strain will cause disease – ColV being one of these virulence factors, the Yersiniabactin High Pathogenicity Island (HPI), which we also identified in the cluster, being another (PMID: 33112851). Outside extra-intestinal sites, ColV plasmids are most likely to be selected in the gut of humans and animals where they provide some fitness benefit (reflected by the abundance of BAP2 ColV+ ST58 sourced from animals - particularly poultry and pigs). Eventually, via pathways that are at present poorly understood, some of these strains encounter extra-intestinal sites where the ColV plasmid contributes to virulence. This concept is introduced in the second to last paragraph of the introduction and covered within the discussion, particularly with updates made in response to this review.

Comment: On the other hand, The authors state that bovines are the primary host of ST58 and in that niche there is a mix of ColV positive and negative isolates. Let's try this another way. Figure 2 presents BAPS4 cluster as having some proportion of ColV positive strains, yet it doesn't seem to be particularly different from the other clusters in any of the figures presented. Perhaps this might be done to the use of the arbitrary definitions of the BAPS clusters. I would be curious if there is a difference between COLV+ve in BAPS4 vs COLV negative. Since they are so closely related. I know the literature supports ColV plasmid acquisition as then conferring some pathogenic potential. But the genomic data presented here just doesn't give enough information to say if this is the case.

Response:

- At no point do we state that cattle are the 'primary host', we state that cattle are the most numerous host from which ST58 was identified in this collection and most of them belong to the BAP6 evolutionary cluster – indicating some sort of association between this subset of ST58, which infrequently carry ColV, and cattle as a host.
- BAP clusters are not arbitrarily defined. They are primarily conditioned on the underlying genomic alignment in conjunction with the tree structure (see Methods and PMID: 31076776)
- We are unsure what is meant by 'particularly different' regarding BAP4, however updated Figures 3 and 7 show that BAP2 is distinct from other clusters with regard to:
 - Amount of ST58 sequences that belong to the cluster
 - The proportion of ColV+ sequences (85% vs 1-44% in the others)
 - Source distributions (notably 33 ExPEC strains in BAP2 vs 1-3 in each of the other clusters)
 - F plasmid types and serotypes
 - Rates of virulence gene and AMR gene carriage (Figure 7- more virulence than all but BAP1, more AMR than all)

Comment: The authors use statements “BAPS3 sequences on average carry significantly more ARGs than other BAP clusters ... ” but then do not quantify significance.

Response: This figure (Fig. 7) and the text have been updated with information on the relevant statistical tests and p-values.

Comment: They state that 16% of ExPEC in their survey have ColV plasmids -- is that significant in some way? No statistical test is provided. No laboratory experiment is provided. No framework for testing this hypothesis is provided. What would be the likelihood of this correlation being observed by random chance?

Response: A stand-alone percentage cannot be judged as significant or not – it is a statistic, based on screening 34,364 *E. coli* genomes with the same marker gene criteria for ColV plasmid inference as was used for the ST58 collection as described in the methods. The data is presented in Figure 8. It is not a hypothesis or a correlation – it is observed genomic data.

Comment: To me, if anything, it appears that IF the BAPS3 clade is truly special (and that is an IF) then it could be some other set of genomic changes and the ColV plasmid is picked up by-the-by, a proxy marker to something happening elsewhere.

Response: We are unsure what is meant by ‘truly special’. The paper clearly identifies other genomic elements that are associated with BAP2 (formerly BAP3) cluster in Figure 6 and addresses these findings in the discussion. The argument seems to be that ColV plasmids are there for no reason at all, which is confounding given the trends in our data and the literature on ColV plasmids. Furthermore, if ColV plasmids were just picked up by-the-by, they would be distributed uniformly across all sources of *E. coli*, across all *E. coli* STs and across the ST58 phylogeny. Our analyses clearly show that this is not the case. ~200kb of DNA is not acquired and retained by *E. coli* for no reason, there is too great a fitness cost for that much redundant genetic material in a ~5-6Mb genome retained for no phenotypic benefit.

Comment: The final sentences of the conclusions regarding the implications of the work are not supported by the results. For instance, detecting ARGs in particular *E. coli* does not mean that that resistance will lead to increase in clinical cases and be “Deleterious to human health” and there are not a large number of human associated ST58 so it does not seem to be an emerging pathogen.

Response: We did not claim that AMR would lead to increased clinical cases – we stated that the association of AMR with ColV could lead to greater complications due to resistance in ColV+ strains that cause infections.

Overall, we have made edits to the discussion and conclusions refining, and in some cases removing arguments regarding the transmission of ST58 through different

sources. We have instead emphasised the influence of numerous sources and genetic elements in the evolution and expansion of the BAP2 lineage and ST58 in general, stressing the need for more human origin strains to address the link between clinical cases and food animals that carry ST58.

Please see the following cited papers that directly refute the assertion that ST58 is not an emerging pathogen:

- The proportion of bloodstream infections caused by ST58 in Paris has doubled in the last twelve years (PMID: 33952335).
- Despite the profound bias towards ESBL+ E. coli in ExPEC literature, ST58 which displays low levels of ESBL carriage according to our analysis is still one of the top 20 global ExPEC STs, and the only phylogroup B1 (PMID: 31189557)
- Also see references 4-12 cited in the introduction that clearly identify ST58 as a globally disseminated human pathogen.

The lack of publicly human and ExPEC origin ST58 in our collection is surprising given the above reports and is discussed in the 'Limitations' section of the discussion.

Reviewer #2 (Remarks to the Author):

The paper of Reid et al. reports a phylogenomic analysis of a panel of 752 E. coli B1-ST58 strains, a clone emerging in animal and human pathologies while becoming resistant to antibiotics. The topic is of interest, although not new. The work is well done and the results are discussed fairly. I have however two major concerns.

Comment: First, the ST58 according to the Warwick scheme corresponds to the ST24 and 87 according to the Pasteur Institute scheme. Together with the ST155 (Warwick nomenclature), they constitute the CC87 (Pasteur nomenclature). The emergence of this clonal complex has been first published in 2015 (PMID: 26613786 -) with several phenotypic tests including conjugation assays that should be discussed. Its recent increase in human bloodstream infections is clearly noted in a 12-year period (PMID: 33952335). There is also an important literature by the Bonacorsi group on the pS88/pS88-like plasmid that should be discussed (see for example PMID: 24086343). In sum, please position your work according to the global knowledge.

Comment: Thank you for bringing these additional publications to our attention. They have all been integrated into the introduction and the discussion to support our arguments that ST58 is an emergent pandemic ExPEC and the role of ColV in virulence.

Comment: Second, I am not convinced by the title and more precisely that it is the ColV plasmids that drive the evolution of the lineage. Pathogenicity in E. coli (excepting perhaps Shigella) is a multifactorial phenomenon. I think numerous other genes (as those of the HPI) are probably involved in the process.

Response: We agree with this comment and have altered the title as well as elements of the discussion to reflect that the evolution of this cluster of ST58, whilst undoubtedly influenced by ColV, is indeed multi-factorial.

Reviewer #3 (Remarks to the Author):

Reid et al describes the detection and characterisation of ColV plasmids in E. coli ST58 from global collection of isolates. The genomic analysis of the isolates are extensive and comprehensive. However, there are some concerns regarding interpretation, relevance, novelty of the findings and number of conclusions that cannot be substantiated with the current study design and epidemiological data.

The study is largely observational with large number of ST58 isolates.

Comment: The major finding from this manuscript is that large number of ST58 carry ColV plasmid. This has been already reported by the authors in a number of recent publications (McKinnon et al 2018 Int J Antimicrob Agents 52(3):430-435; Wyrsh et al Microbial Genomics 2020;6; Reid et al 2020 Front. Microbiol;)

Response: This paper is a significant advancement from those works which present sporadic evidence of ColV in ST58. This paper clearly builds on a significant scale by including hundreds of genomes and not only demonstrating that ColV plasmids are common, but they are in fact characteristic of a specific evolutionary group of ST58; a group which is frequently isolated from food production animals and carries numerous other characteristic genetic elements. None of this was identified by those previous papers.

Comment: There is no experimental in-vivo study undertaken to clarify the virulence of these strain/ plasmid using murine model. Which is one of the common approaches taken to clarify virulence of ExPEC isolates

Response: Whilst in-vivo data is always desirable, this was outside the scope of the paper. We set out to perform a genomic investigation based on a) a wealth of literature demonstrating ST58 is an increasingly common pathogen, and b) the known contribution of ColV plasmids to pathogenicity in a variety of genomic backgrounds.

Comment: ColV has been well characterised (Johnson TJ J Bacteriology. 188, 2) as a potential virulence plasmid and ST58 has been previously reported as a potential animal pathogen and it is not surprising that the strain carry ColV plasmid. (Fuentes-Castillo et al Transboundary and Emerging Disease).

Response: Respectfully, I think it is obvious that this manuscript goes a long way beyond those two observations.

Comment: Coverage biases in Illumina sequencing and its impact on quality of data and inferences are not addressed

Response: Researchers in the field should be aware of the strengths and weaknesses of Illumina sequencing - it is considered a gold standard for short read sequencing of bacterial genomes. Coverage bias in microbial sequencing specifically, typically affects sequencing of organisms with unbalanced GC content, however *E. coli* is typically ~50% GC and only approximately 0.058% of the *E. coli* genome contains GC-rich ($\geq 75\%$ GC) sequences (PMID: 34166413). It is unlikely to have affected any inferences made in this paper. As for data quality, all sequencing reads have been quality controlled either by SRA or via Enterobase upload as described in the Methods, furthermore we have now added assembly statistics to the metadata in Table S1 to support the quality of the assemblies used in the study.

Comment: A number of isolates carried ESBL genes, could the over representation of ST58 in the study could be attributed to antimicrobial resistance rather than virulence of this isolates. This has been the case for uropathogenic *E. coli* ST131, a moderately virulent strain becoming a pandemic ExPEC strain due to the carriage of fluoroquinolone and/third generation cephalosporin resistance?

Response: Over-representation of ST58? This whole study was selected based on the strains being ST58. We clearly state and discuss in the paper how ESBL carriage was low and not localised to any phylogenetic cluster and therefore unlikely to explain the expansion of BAP2 or any other lineage of ST58. If this were anything like the case of ST131-*H30*, near ubiquitous ESBL carriage localised to specific clades would be observed.

Comment: What is the fluoroquinolone resistance status of these isolates, that may explain it emerging status? This has also been seen in other emerging ExPECs such as *E. coli* ST648 (Schaufler et al Antimicrobial Agents and Chemotherapy 2019 63, 6) and ST1193 (Johnson et al 2019 Journal of Clinical Microbiology 57, 5; Johnson et al Antimicrob Agents Chemother 2018 Dec 21;63)

Response: We do not have phenotypic fluoroquinolone resistance data however SNPs that typically confer this sort of resistance were only present in 9% of strains and were distributed across clusters at rates of 1-19% per cluster (12% in BAP2), FQR is therefore highly unlikely to explain the expansion of BAP2 or differentiation of any other cluster. We have added this data to the results and Fig. S8.

Comment: Could ST58 be a dominant animal associated clone and a sub population of humans may acquire this clone either by direct transfer through the environment/ food as indicated by Borges et al 2019 Foodborne Pathog Dis . 2019 Dec;16(12):813-822.

Response: Absolutely. Our data supports this argument.

Comment: ST58 has also been reported to as an atypical EPEC (Jouini et al Antibiotics 2021, 10, 670). How many of the isolates in this study are aEPEC. This need to be clarified.

Response: The intimin gene *eaeA*, characteristic of aEPEC (in the absence of *bfp*) was not identified in the study.

Comment: Figure 7 is not informative. Carriage of ColV plasmid naturally would increase the putative virulence genes scores. In addition, biases in isolates used in this study (virtue of convenient sampling) could result in difference in carriage of virulence/ antimicrobial resistance genes.

Response: We agree that it is obvious that the virulence score would increase for ColV+ relative to ColV- negative in Fig. 7d but disagree that this makes the figure uninformative. The figure also shows how the BAP2 differs from other clusters with respect to VAGs and ARGs (a) and b)) and the reflection of this trend in ColV (c) and d)) serves as a probable explanation for both phenomena.

Convenience sampling public genomic datasets is profoundly different from convenience sampling bacteria from one's immediate environment, which might be expected to display significant bias in VAG and ARG distribution. Had we sampled only one source from one geographic area – of course we would expect the ARG and VAG results to suffer from sample bias, however this is not what we have done. Given the global distribution, wide variety of sources and absence of antimicrobial selection in our collection, we would argue that we have assembled the least biased collection of ST58 possible. Having observed these trends within a clear population structure that doesn't cluster exclusively by geography or source, we consider them to be free from avoidable sources of sample bias.

Comment: Lines 415-423 – Virulence of these isolates are not well characterised; carriage of colV alone does not make the strain highly pathogenic. This is evident in the meta analysis where ST58 was a lower ranked ExPEC (ref 2; Manges et al 2019 Clin Microbiol Rev 32)

Response: As this was a genomic study, it was not feasible to phenotypically characterise the virulence of the 752 strains. However, I believe our observations in conjunction with the literature and discussion points support the innate virulence of BAP2 strains, notwithstanding host factors. The meta-analysis by Manges et al did not rank ExPEC STs based on innate virulence as this comment seems to imply; the ranking of each ST reflected the frequency at which it is observed in extraintestinal infections by inference from its prevalence in published literature. The fact that ST58 appeared in that meta-analysis at all indicates that it has been repetitively observed as an actual pathogen and our study has identified genomic factors that provide some explanation for that observation.

Comment: Lines 493-507 Isolation from cattle does not mean cattle adapted. The conclusions drawn are weak and not substantiated with the limited data set used in this study

Response: This section has been reworded to reduce the speculation around sources of transmission. Nonetheless, it is highly likely that a subset of ST58 are adapted to cattle given the dominance of cattle-derived strains in the second largest evolutionary cluster of ST58 (BAP6). Furthermore, ST58 was also previously identified as a predominant lineage in faecal *E. coli* of cattle (PMID: 31439690).

Comment: Lines 510-527. The conclusion on vectors-sources of CoIV is not substantiated by this study. There is no experimental evidence from this study or other published literature make this conclusion.

Response: We have removed this section as we agree it was too far removed from the actual data in this study, however these aspects are always within our thinking regarding the transmission of pandemic lineages of *E. coli* through different sources.

Comment: Lines 442-454. A large number of studies have reported *iroN* and other siderophore systems among large number of *E. coli* strains therefor CoIV and *iroN* unlikely to be associated with grazing resistance

Response: We disagree that the distribution of these systems across the species precludes clear evidence of their involvement in protozoan grazing resistance as described by Adiba *et al* (PMID 20711443). Nonetheless, this section has been removed in favour of discussion of the role of iron acquisition systems in gut fitness more broadly.

Comment: It is unclear why CoIV carriage in humans faecal commensal is urgently needed? CoIV has been reported previously and how that information may prevent humans from getting extraintestinal infections is unclear.

Response: The human gut is the pre-infectious reservoir of ExPEC that infect the urinary tract (PMID: 31797927, PMID: 25853778, PMID: 26542041, PMID: 9072556). If CoIV plasmids are significant contributors to these infections, then knowledge of their carriage in the gut could contribute to many aspects of research that might negate their impact on public health.

Comment: A number of statements are not backed up by references example Lines 415-423

Response: References added.

Comment: Lines 542-545. It is unclear how plasmid ecology could assist with risk ranking of bacteria. Perhaps carriage of resistance and virulence assist but just the

carriage of plasmids may not help. See Collineau et al Front Microbiol. 2019; 10: 1107 for approaches undertaken for quantitative risk assessment using WGS for food AMR.

Response: We were not referring to empirically calculated risk, more the general idea of risk posed by plasmids that circulate in non-clinical sources yet are relevant to instances of infection. We have decided to remove the word to avoid confusion. Of course, one would not just consider carriage of plasmids in isolation – one would **always** simultaneously consider the functions they encode such as virulence, AMR, fitness, metabolism etc and how their distribution throughout different host and environments may have relevance to pathogenesis and public health.

REVIEWERS' COMMENTS

Reviewer #1 (Remarks to the Author):

The authors have addressed my comments and have been patient in explaining my misunderstandings.

Thank you. I look forward to seeing it in print!